# The Cluster Variation Method: A Primer for Neuroscientists

**DOI:** 10.3390/brainsci6040044

**Published:** 2016-09-30

**Authors:** Alianna J. Maren

**Affiliations:** Northwestern University School of Professional Studies, Master of Science in Predictive Analytics Program, 405 Church St, Evanston, IL 60201, USA; alianna.maren@northwestern.edu; Tel.: +1-703-618-5847

**Keywords:** brain–computer interfaces, pattern recognition, statistical thermodynamics, Cluster Variation Method, entropy, brain networks, functional motifs, variational free energy, neural activation patterns, deep learning

## Abstract

Effective Brain–Computer Interfaces (BCIs) require that the time-varying activation patterns of 2-D neural ensembles be modelled. The cluster variation method (CVM) offers a means for the characterization of 2-D local pattern distributions. This paper provides neuroscientists and BCI researchers with a CVM tutorial that will help them to understand how the CVM statistical thermodynamics formulation can model 2-D pattern distributions expressing structural and functional dynamics in the brain. The premise is that local-in-time free energy minimization works alongside neural connectivity adaptation, supporting the development and stabilization of consistent stimulus-specific responsive activation patterns. The equilibrium distribution of local patterns, or *configuration variables*, is defined in terms of a single interaction enthalpy parameter (*h*) for the case of an equiprobable distribution of bistate (neural/neural ensemble) units. Thus, either one enthalpy parameter (or two, for the case of non-equiprobable distribution) yields equilibrium configuration variable values. Modeling 2-D neural activation distribution patterns with the representational layer of a computational engine, we can thus correlate *variational free energy minimization* with specific configuration variable distributions. The CVM triplet configuration variables also map well to the notion of a M = 3 functional motif. This paper addresses the special case of an equiprobable unit distribution, for which an analytic solution can be found.

## 1. Introduction

The purpose of this paper is to introduce neuroscientists to cluster variation methods and their potential application in systems neuroscience. This primer focuses on brain computer interfacing; however, the contribution is more wide ranging and can be summarised as follows: If distributed systems (like the cortex) can be characterized as a two-dimensional lattice with local coupling, there exists a set of mathematical tools that enable one to quantify the distribution of local interactions. In brief, these (cluster variation) tools allow one to prescribe the distribution of local configurations in terms of a single parameter; namely, an interaction enthalpy. This plays a similar role to the temperature in statistical thermodynamics, thereby prescribing a distribution over configurational variables. This formulation has a number of potentially exciting applications. For example, under the assumption that biological systems minimize their free energy, the ensuing equilibrium distribution of local configurations can be determined for any given interaction enthalpy.

This offers a principled and parsimonious way to specify prior probability distributions over the distributed states of (neuronal) networks. These prior distributions may play an important role in either generating random networks for statistical comparison in graph theory, or in furnishing observation or generative models for empirical data via prior constraints. Another application of this technology would be to summarize any empirical distribution of local configurations in terms of a single parameter, enabling one to test for differences between different brain states or, indeed, different subjects and diagnostic categories. Finally, the ability to summarize the statistics of local configurations on lattices paves the way for testing key hypotheses about network organization; for example, do real brain networks minimize free energy for a given interaction enthalpy? We will return to some of these ideas in the discussion.

A major challenge in constructing effective *Brain–Computer* Information *Interfaces (BCIIs)* is the difficulty in not just accurately classifying time-varying neural activation patterns, but in creating accurate predictions. Bayesian prior probabilities have often been used for this task [1,2]. However, it can sometimes be difficult to compute these prior probabilities, especially considering the growing complexity of BCI data. One means of obtaining the needed Bayesian priors is to use *Bayesian inference*; for example, *approximate Bayesian computation*, which yields the desired values through *variational free energy minimization* [3,4].

Applying this approach to BCIs requires the identifcation of a computational substrate whose behavior can be described using statistical thermodynamics. In particular, we need to define a system which can be characterized via various free energy-minimized states, so that variational free energy minimization can model perturbations to these states.

Using a classic Ising model to characterize such a computational substrate has various limitations [5]. In particular, the classic approach yields only limited information, which is insufficient to describe the pattern-richness of neural activations. A more interesting approach is the use of the cluster variation method (CVM) [6,7], which specifies the distribution of not only simple on/off (or **A**/**B**) units, but also the distribution of local patterns (nearest-neighbor, next-nearest-neighbor, and triplets).

The distribution of local triplet patterns (where triplets comprise the *basic cluster* for this formulation) can be of particular interest to brain researchers, as substantial work indicates that groups of three units each (functional and structural motifs which have a Motif number or M = 3) are a minimal computational unit for describing fundamental brain connectivity dynamics.

Up until now, the CVM has been largely unknown to brain researchers. This paper attempts to serve the brain research—and particularly the BCI—communities by offering a tutorial-style CVM introduction. The specific values and unique values of this paper to these communities are:
Correspondence between CVM triplet configuration variables and the M = 3 functional motif (building block of three neural units) provides a natural means for interpreting CVM-based results in terms of functional motifs in the brain and vice versa,Local-in-time free energy minimization can potentially help stabilize specific neural learned response patterns, andCVM modeling bridges local pattern distributions to other important realms, including brain activity at critical states, the generation of Bayesian belief probabilities, complexity, and other factors.

Structural and functional motifs are a means of expressing the characteristic building blocks in neuroanatomical data. Functional motifs of M = 3 (a set of three completely-connected neural units) represent a local pattern configuration that both maximizes the number of local patterns that can be achieved while simultaneously minimizing the number of structural motifs needed to support the diversity in functional patterns [8].

The CVM ergordic distribution can be specified using either one enthalpy parameter *h* (for the case of equiprobable bistate unit distribution) or two parameters (when that constraint is released) [9,10]. Thus, selecting a single parameter value (for equiprobable unit distributions) specifies a distinct point in the trajectory of equilibrium-stabilized (or non-equilibrium steady state for open systems) *distributions* of local patterns.

The interaction enthalpy parameter *h* corresponds to an average pairwise coupling strength between neural units. As will be discussed in Section 2.2, several successful models for neural organization and dynamics are built using predominantly pairwise connections. As an illustration, Tkačik et al. have combined experimental and theoretical studies to propose specific values for the pairwise interaction enthalpy in thermodynamic models of (vertebrate retina) neural systems which closely emulate actual neural dynamics [11,12]. They further note that their model “accurately predicts the correlations among triplets of neurons”, providing a further rationale for applying the CVM with a basic cluster of triplets to neural organization [11].

Characterizing the distribution of local patterns, we also have a new means for interpreting the kind of information present in a 2-D ensemble. When the enthalpy parameter *h* is chosen to yield greater diversity in local pattern distributions, more information can be encoded. Additionally, because the CVM approach works with distributions of triplet patterns, this method usefully connects with means for the characterization of functional motifs where M = 3 [8].

Additionally, work by Friston et al. (2012) shows how Bayesian priors can be derived as outcomes of variational free energy minimization [13]. This specifically addresses notions of self-organization within the brain. While Friston et al. postulate how free energy minimization can be intimately tied to a range of important dynamic phenomena in the brain, the next step is to represent the free energy associated with spatio-temporal neural activity *patterns*. Thus, a means for explicitly representing pattern distributions can be most useful.

The CVM also has potential as a computational substrate within deep learning architectures, addressing the needs of generative methods to model diverse patterns over time.

In short, the Cluster Variation Method (which has received substantial attention in areas such as materials science) has potential application for BCIIs, both in supporting direct models of neural activity, as well as providing a computational substrate for an adaptive pattern recognition engine.

Because the CVM formalism is not yet widely known, this paper includes both tutorial and review aspects. The following Section 2 describes both the challenges and the overall context for the creation of next-generation BCIIs, together with identifying some key related challenges in understanding neural activity as well as neural structure and function. Section 3 briefly overviews how statistical thermodynamics already plays a role in modeling neural activity, from relatively small ensembles up through global networks, and identifies key methods and experimental results (both computational along with in vivo and in vitro studies) for which statistical thermodynamics has already played a major role.

Section 4, Section 5 and Section 6 introduce the Cluster Variation Method itself. Section 4 introduces and describes the *configuration variables*, which are used to characterize local patterns. It provides a very small but useful illustration of how pattern distributions correlate with the interaction enthalpy parameter *h*, using a 1-D CVM to facilitate an immediate intuitive understanding.

Section 5 provides the theoretical framework for the 1-D CVM, again providing an instance within intuitive reach. Section 6 builds on this, providing the theoretical framework for the 2-D CVM, which will typically be more appropriate for modeling neural systems as well as for a BCII computational substrate.

Section 7 continues with a focus on the 2-D CVM, overviewing next steps connecting the 2-D CVM with both anticipated results for the modeling of neural-like systems with a “rich-club” topography and implications for a computational engine that can be used to interpret neural activation patterns. Section 8 provides concluding remarks.

## 2. Background

The rapidly-advancing field of Brain–Computer Interfaces (BCIs) is undergoing a major transition as we move from relatively sparse sensor assemblies (typically external to neural tissue) towards highly dense intra-cortical sensor arrays. As an illustration of this transition, Stevenson and Kording have published summary data showing that the number of neural recording electrodes has doubled approximately every seven years, over the 1960–2010 timeframe [14,15].

Three key areas rapidly hastening the onset of effective intra-cortical BCIs include developments in:
Electrodes and other implantable materials with reduced neuro-inflammatory responses, including recent advances in carbon nanotube (CNT) electrodes [16,17,18,19,20,21,22],Novel fabrications allowing much greater sensor density [23,24,25,26], andImproved wireless communication and power transmission technologies [27,28,29].

We are thus realizing ever-increasing granularity and refinement in sensed neural activations, in both the spatial and temporal realms. As Tognoli and Kelso state, “From a viewpoint that separates observable and true dynamical properties of neural ensembles, state transitions observed in EEG (electroencephalography) and LFP (local field potential) recordings (STobs) cannot be taken at face value, since they are the indirect manifestation of the dynamical organization of their originating neural ensembles. The issue can be turned the other way around, however, by examining which neural organizations appear as state transitions in neurophysiological recordings” [30].

However, even with externally-placed sensors, increased sensor density is encouraging a greater range of applications, ranging from entertainment to the detection of order–disorder transitions in the brain [31,32]. Maren and Szu describe how a CVM-based approach to modeling EEG signals can be used in a dry-mounted commercially-available EEG recording system, suitable for daily use [33].

Neural activation patterns are related to functional connectivity within the brain, and functional connectivity is in turn related to and dependent upon the underlying structural connectivity. Recent work has established that functionally-connected neural groups are indicated by both search information factors as well as path transitivity along shortest paths—both of these leading to improved communications [34].

In addition to (and potentially complementary to) the methods already advocated, there is another means by which functionally-effective neural patterns can be characterized. This method again fosters a high degree of information storage and communications pathways. This complementary approach uses statistical thermodynamics, particularly the Cluster Variation Method (CVM) [6,9,10,33].

There is an entirely different, although related, role for CVM in modeling brain activation patterns for BCI. This role is that—in addition to directly modeling neural activations themselves—we need a computational system which functions as a pattern repository and classification engine, operating in concert with the time-varying patterns detected in the brain. For this related role, the most common approach thus far has been to construct various categorization engines, interpreting real-time neurally-generated signals [35]. Most significantly, recent advances in deep learning [36] build on earlier neural network architectures [37,38,39] and use brain architecture-based principles to solve very computationally tough problems.

Thus, this paper introduces the Cluster Variation Method for use in Brain–Computer *Information* Interfaces (BCII), with two purposes: (1) Identifying how the CVM functions in a complementary and supportive manner to existing and ongoing work on functional and structural motifs, dynamics of neural ensembles, and the characterization of neural pattern formation, providing additional input to a BCII system; and (2) Providing the basis for a new BCII computational engine, along the lines of successful deep learning generative architectures, invoking equilibrium processes to stabilize pattern formation in response to specific stimuli. In short, the next demanding challenge deals not so much with the interface itself, but with the *information* interface, the Brain–Computer *Information* Interface (BCII).

The background given in the following three subsections addresses three distinct areas:
Means for decoding the information content in neural pattern formation and activation,Pairwise and longer-range neural connectivity, andFunctional and structural motifs, neural topographies, and pattern formation: structural and functional organization of small units within the brain (or *motifs*), as they can influence BCII signal classification.

### 2.1. Decoding Information in Neuronal Activation Patterns

The following two subsections address two components of the Brain–Computer Information Interface:
Modeling the signal itself, which will continue to be a challenge in what will still be a low signal-to-noise environment, andUse of Bayesian statistics for the classification of neuronal outputs, identifying both current applications and important challenges in the extension of this approach.

#### 2.1.1. The Recurrent Quantum Neural Network

Gandhi et al. have advocated an interesting approach to modeling the time-varying signals produced by a neural ensemble by using a recurring quantum neural network (RQNN) [35], which further develops an approach introduced by Behera et al. [40]. They have applied the RQNN to classifying EEG signals corresponding to subject’s motor imagery (MI) of a desired task [41].

A key factor in the RQNN approach is that the average behavior of a neural lattice that estimates the EEG signal is a time-varying probability density function (*pdf*). Each neuron within this computational lattice mediates a spatio-temporal field that aggregates the *pdf* information from the observed noisy input signal. The RQNN approach enables the online estimation of the *pdf*, which then allows the estimation and removal of noise from raw EEG signals, thus addressing a key challenge in working with such low signal-to-noise signals.

The RQNN can be viewed as a Gaussian mixture model estimator of potential energy with fixed centers and variances, with variable weights in the neural network component of the model. Gandhi et al. found significant improvement in RQNN results compared with both the well-known Savitzky–Golay (SG) filter as well as the Kalman filter. They note that one particular advantage of the RQNN method is that, unlike the Kalman filter (for which the statistical behavior of noise and signal are assumed), the RQNN makes no such assumptions. Rather, the RQNN directly estimates the *pdf* without making such assumptions, and can thus preferentially enhance the EEG signal, as the noise *pdf* is naturally non-Gaussian. Thus, the RQNN may be a valuable tool for improving EEG signal enhancement. It is likely that the same method would potentially be extensible to extracting meaningful signals from more granular neural ensembles.

#### 2.1.2. Bayesian Methods

A primary goal with BCI is to infer intention, given neural inputs. These may be relatively stationary tasks (e.g., classification) or smoothly-varying over time (e.g., motor control). Very often, Bayesian methods are crucial to the interpretation of either or both neural signal correlates and/or task intentionality [1,42,43].

Bayesian methods also play a role in the information-processing component of BCIs; that is, within the Brain–Computer *Information* Interface (BCII). For many applications, classic information theory methods suffice [44,45,46]. However, these methods typically rely on the ability to model a distribution, or a set of responses to a given stimulus, in terms of prior history. It is not always feasible to obtain these conditional probabilities, or to rely on them once they have been collected. In addition, neural systems are adaptive; they can change their responses over time [47,48,49]. Another factor is that as we expand to more practical and pervasive interfaces, there are likely to be more extraneous stimuli and internal distractions. These will not always readily fit into a predetermined modeling schema.

Thus, Bayesian models are likely to be essential as we create the next level of BCIIs. However, we will need to expand on the basic methods. In particular, we will find it useful to approximate Bayesian priors. Work in approximate Bayesian inference will be useful [3,4,50].

### 2.2. Pairwise Correlations Crucial to Neural Pattern Formation

Our understanding of both the functional and structural connectivity of neurons and local neural groups has evolved well beyond the initial ideas proposed by Hebb [51]. An early important influence was the work of Edelman in proposing cortical organization with dynamic re-entry [52,53]. Singer’s work led to early appreciation of long-range cortical activity synchronization [54]. A careful review by Honey et al. [55] advances theoretical arguments for how the brain’s network topology and spatial embedding should strongly influence network dynamics.

One of the most powerful architectural themes of the neocortex is that of columnar organization, as described by Edelman and Mountcastle in *The Mindful Brain* (1978) [52], and particularly by Hubel and Wiesel regarding the visual cortex [56,57]. As Stevenson et al. note [2]: “One of the primary hypotheses about network structure is that neurons, especially cortical neurons, form groups or assemblies characterized by similar stimulus–response properties... [referring] not only to localized cortical columns, but also to functional assemblies that may be more spatially distributed.”

Stevenson et al. show how functional connectivity between neurons can be identified using Bayesian priors in addition to point spike observations, further building a model-based approach to the identification of “effective connectivity”, as introduced by Aertsen et al. [58]. Their analysis uses two suppositions about neural interactions (supported by their results, together with work by others): (1) neural connectivity is sparse, even within small functional ensembles where the same neurons may have similar response properties; and (2) the influence of one neuron on another varies smoothly over time.

Fries suggests that neural coherence mechanistically subserves neural communication [59]. With this notion, there would be an advantage to training neural connections as well as firing patterns that unify activations and responses across a set of communicating neurons. Mechanisms of neural adaptation and learning, leading to activation coherence, could serve an information theoretic (or communication) purpose.

Along these lines, neural coherence or mutual activation patterns can occur at different timescales and spatial ranges, contributing to nested or hierarchically-embedded behaviors. Tognoli and Kelso advocate a theory of Coordination Dynamics (CD), by which interactions occurring at different strengths and persisting over different timescales can lead to diverse metastable states [30].

Recent studies of both structural and functional connectivity have uncovered several topological themes, including “rich clubs” in which hubs are densely interconnected in a “hub complex” [60,61]. These hub complexes are a set of hub regions that are more densely interconnected than predicted by chance alone [62]. This extends the previously-known topographies, such as small-world and (to a limited extent) scale-free networks [63]. As Alagapan et al. have noted, hub-like structures promote information transmission within neural assemblies [64]. Such complex topologies can be supported by diverse intrinsic coupling modes, which provide multiscale ongoing connections between neural groups, even in the resting state [65].

Although there are indeed long-range neural connections, functional groups of neurons can be induced via multiple pairwise connections. As Schneidman et al. note, “small correlations among very many pairs could add up to a strong effect on the network as a whole”, and they identify an Ising-like minimal model [66] which captures properties of large-scale networks [67]. Cohen and Kohn likewise review measures of correlation for pairwise connections between neurons, focusing on the “measure of the degree to which trial-to-trial fluctuations in response strength are shared by a pair of neurons” [68]. In a related line of work, Poli et al. have used a partial correlation approach to studying how structural connections underlie functional-effective ones in dissociated neuronal cultures, finding that the partial correlation approach is beneficial in determining both structural and topological features [69].

The significant role of pairwise interactions is further supported by Alegapan et al. [64], who have recently identified convergence and divergence properties of neurons, contributing to the formation of what Sporns et al. have described as a “connectome” [70], noting that the “success of a model that includes only pairwise interactions provides an enormous simplification ...”

These arguments lead us to think that a statistical thermodynamics approach that combines pairwise interactions with the modeling of very local clusters will be useful. One crucial role of such a statistical thermodynamics-based model would be supporting coherent activities within neural ensembles. As Zalesky et al. comment on time-resolved resting state brain networks [71], “This result verifies that nonstationary fluctuations are synchronized across functional brain networks in such a way that multiple connections transition *en masse* ... [so that] dynamic fluctuations in functional connectivity at the pairwise level appear to be coordinated across the brain so as to realize globally coordinated variations in network efficiency over time...”.

Thus, it is reasonable to assume that a statistical thermodynamics model of the Ising-type, that is, invoking pairwise correlations, will be a useful starting point. The crucial step proposed in this work is that modeling cluster behaviors—that is, distributions of local patterns—will also be useful.

### 2.3. Neural Motifs, Topographies, and Pattern-Formation

The question that naturally arises is thus: How extensive should the local pattern models be? That is, even if we restrict our neural interaction models to pairwise interactions, do we need to model large local patterns to effectively model neural organization, or will very small local patterns suffice?

This is an important question, as Sporns and Kötter hypothesize that “... the connection patterns of real brain networks maximize functional motif number and diversity, thus ensuring a large repertoire of functional or effective circuits, while they minimize the number and diversity of structural motifs, thus promoting efficient assembly and encoding.” They further suggest that “brain networks maximize both the number and the diversity of functional motifs, while the repertoire of structural motifs remains small” [8].

An answer comes, at least in part, from studies identifying that neural motifs (or functionally- and structurally-connected units) on the order of M=3 (where *M* is the number of units) is a useful base. As noted, again by Sporns and Kötter: “Particularly interesting is the increased occurrence of a single motif at *M = 3* ... and its expanded versions at M=4 ... this motif type combines two major principles of cortical functional organization, integration and segregation ...”. As corroborated by Papo et al., “Brain graphs also have higher clustering than random graphs. For any graph, clustering can be quantified by counting the proportion of triangular motifs that completely connect a subset of three nodes” [72].

Out of all this work comes a leading role for the clustering of neurons and neural groups within the brain. In particular, Stevenson et al. note that “clustering is not trivial; that is, it is not purely random or a result of bad spike sorting,” and they hypothesize that “neurons in the same clusters may have similar relationships to external covariates, that is, similar common input” [2].

One important area of investigation that will have a strong impact on BCII will be the influence that structural connectivity has on functional organization. Gollo and Breakspear [73] identify two crucial features that shape cortical dynamics: the numbers of resonant pairs (within an ensemble) and frustrated motifs, respectively. They find that the strength of the coupling between pairs plays a crucial role: too strong a coupling, and the neural ensemble becomes globally connected; and too weak a coupling reduces variability, because too few pairs can become synchronized. They note that “spatial clustering leads to a modular architecture, in which nodes are strongly interconnected with regions within the same cluster (SC)—typically nearby in space—than with regions outside their cluster”.

Similarly, Gollo et al. have established the crucial role of resonance pairs within cortical motifs, noting that “the presence of a single reciprocally-connected pair—a *resonance pair*—plays a crucial role in disambiguating those motifs that foster zero-lag synchrony in the presence of conduction delays (such as dynamical relaying) from those that do not (such as the common driving triad). Remarkably, minor structural changes to the common driving motif that incorporate a reciprocal pair recover robust zerolag synchrony” [74].

We can summarize the discussions of these last two subsections by noting that:
Although long-range (inter-cluster) correlations play a significant role, a model focusing on pairwise correlations (including the notion of resonance pairs) is a good starting place, andRelatively small functional units or *motifs* (i.e., M=3, with correspondingly small structural units) play a strong role in neural activation patterns.

We go forward into investigating models with two minimal conditions:
Pairwise interaction modeling is essential, and although longer-range correlations are important, we can potentially create a starting point that focuses on local pairwise correlations, andCluster modeling should be included, and relatively small clusters (M=3) will suffice for correlation with valuable neurophysiological insights.

The following Section 3 overviews how various models—drawn from statistical thermodynamics—are useful for modeling neural activation patterns.

## 3. Statistical Thermodynamics for Neural Ensembles

Over the past several decades, statistical thermodynamics methods have grown in importance for modeling a wide range of neural dynamics and activations, at levels ranging from small ensembles up to large-scale behaviors [75]. Such methods have particularly been useful for modeling critical state phenomena in the brain, although many other applications have also been found.

The most well-known statistical thermodynamics method applied to neural ensemble modeling has been the classic Ising spin glass (Table 1). This has been used with success in several areas. Barton and Cocco, in particular, have used Selective Cluster Expansion (SCE) to model clusters in both cortical and retinal fields [76].

However, one of the greatest constraints in using the simple Ising spin glass approach is that the entropy formulation addresses only the distribution of units into one of two states: active and inactive (or **A** and **B**). Recent developments in neural activation modeling address the formation of local patterns (discussed in the previous Section 2), such as are found with functional motifs. Thus, it is appropriate—and by now, necessary—to move beyond the simple Ising formulation.

Fortunately, there is a means for modeling a system’s free energy that includes local pattern formation: the *Cluster Variation Method*. This method directly invokes not only whether or not a given neural unit is active or inactive, but also the local configurations of such units. Note that these “neural units” are not size-constrained; they represent the building blocks for pattern-modeling within an ensemble. As such, they may be cortical columns, or local areas identified by a parcellization method, whether volume or surface-based [71]. It is this process of modeling local patterns via *configuration variables* that gives us a new tool for the characterization of not so much specific patterns, but rather pattern distributions within large ensembles. Presaging a topic that will be addressed more completely in following sections, there is an important role for this kind of advanced entropy modeling in BCI-related pattern recognition.

One of the key findings that will be presented in the following sections is that the distribution of local patterns (described via *configuration variables*) is a function of enthalpy parameters. In particular, for the specific case where $x_{1}=x_{2}=0.5$ (equiprobable distribution of units into states **A** and **B**, which may be thought of as active and inactive neural units), there is an *analytic solution* that gives the equilibrium values for the configuration variables in terms of a single parameter: the interaction enthalpy parameter *h*.

Although this paper deals exclusively with the case of equiprobable distribution of units into states **A** and **B**, it is useful to briefly mention how non-equiprobable **A** and **B** distributions would evidence themselves in terms of CVM free energy minima (stable solutions). These remarks are being given in advance of a more detailed manuscript which would have to follow this one, in which the specific effects of non-equiprobable distributions could be examined in detail. In brief, though, the impact would be as follows. First, there would need to be two enthalpy parameters, $h_{1}$ and $h_{2}$. One of these ($h_{2}$) would correspond to the single interaction enthlapy parameter *h* as we have been using it in this discussion. The other ($h_{1}$) would be a measure of the enthalpy difference between the two states **A** and **B**. Second, the free energy minima (stable states) would have to be determined computationally, as the analytic solution presented here would not extend to the nonequiprobable case.

In advance of these computations, though, we have some sense of how the parameter values would impact unit distributions and cluster formation. Specifically, the more that $h_{1}$ diverges from zero, the more that the population of neural units will be shifted into non-equiprobable values. That is, there would be considerably more units in one state versus the other.

Nevertheless, even in the case of non-equiprobable unit distributions, the fundamental influence of the $h_{2}$ interaction enthalpy parameter would be consistent with what we will observe in this paper; higher values of $h_{2}$ will encourage the formation of like-with-like clusters, and lower values will correspond to like-near-unlike nearest-neighbor pairings, which will result in a much lower appearance of like-near-like clusters.

Since the influence of the $h_{2}$ interaction enthalpy parameter will be consistent (whether or not $h_{1}=0$), this paper serves the role of introducing cluster formation and characterization (appropriate to modeling triplets of neural units), regardless of the proportionate amounts of active and inactive units.

In short, the ergodic distribution of local patterns (the distribution that would be observed if carried out over a sufficiently large sample, allowed to stabilize over a sufficiently long time) corresponds to either a one- or two-parameter enthalpy specification. This paper focuses on the particular case of equiprobable distributions, for which a single (interaction) enthalpy parameter is sufficient.

The value of this for BCI applications is twofold:
Directly modeling neural activation patterns; this characterizes local activation patterns across an ensemble, together with providing a simple means of modeling the time-evolution of local pattern distributions, andAs part of a pattern recognition engine interpreting neural activations, it provides a one- (or two-) parameter high-level pattern characterization, as well as a potential predictive component that can be interpreted via Bayesian belief probabilities.

A next step would obviously be to correlate the abstract model in *h*-space with specific changes in local neural activation patterns; this invokes correlation with functional networks.

This paper presents only a first step in the direction of what is admittedly a large-scale (and rather ambitious) goal. In order to introduce the method and make clear the mapping from configuration variables to the interaction enthalpy, it focuses exclusively on the case of equiprobable distribution into states **A** and **B**. While this is a substantive constraint, it allows us to work with analytic results and an associated well-defined mapping of equilibrium configuration values to an *h*-trajectory.

The remainder of this section sets the stage for addressing the more sophisticated CVM entropy formulation by summarizing significant related work on three important underlying topics:
Statistical thermodynamics of small-scale neural assemblies (such as neural columns), together with models of larger-scale neural assemblies,Criticality and phase transitions in neural ensembles, and their associated role in maximizing rich information content, andThe role of criticality as it relates to complexity and entropy, providing a rationale for high-entropic neural systems.

### 3.1. Statistical Thermodynamics for Small-Scale and Larger-Scale Neural Assemblies

The most common approach to modeling the statistical thermodynamics of neural collections is the Ising spin glass model, which describes a system composed of bistate units; those which can be either “on” or “off”, or **A** or **B**. This approach leads very naturally to connections with information theory and complexity. The basic Ising model has been applied with some success to collections of neural units ranging from small- to large-scale.

At any scale, models of the statistical thermodynamics of neural systems are a corollary to the more widespread approach of directly modeling neurodynamics, for which the work by Wilson and Cowan is a strong progenitor [77]. Moran et al. have reviewed a set of such models, collectively identified as “dynamical causal modeling” (DCM), which leads to identifiable response states in response to distinct stimuli [78]. These models share the common feature of describing inputs to a given neuron (or neural population) in terms of neural masses and fields. Along these lines, Yang et al. have investigated a four-parameter deterministic Izhikevich neuron model, and have found that it has a wide information transmission range and is generally better at transmitting information than its stochastic counterpart [79].

While dynamic models are valuable, and in some cases lead to stationary or limit-cycle behaviors, the process of free energy minimization can also play an important role. As an example of small-scale modeling, Tkac̆ik et al. have used a maximal entropy approach to model an interacting network of neurons [11,80,81]. They have found that pairwise interactions supplemented by a global interaction controlling synchrony distribution (“K-pairwise” models) extends the more common pairwise Ising models, and provides an excellent account of the data obtained from populations of up to 120 neurons from the salamander retina.

Moving to a larger scale, one of the most valuable contributions of statistical thermodynamics to modeling neuronal systems comes from the investigation of the dynamic properties of activated neuronal ensembles [82]. Indeed, Friston (2010) has proposed that free energy minimization serves as a unifying theory for the description of neural dynamics [83]. Maren has described the phase space for an Ising spin glass system allowing for all-or-nothing phase transitions, coupled with both hysteresis and metastable states, suitable for modeling neural columnar activations [5,84,85].

One of the most important recent findings is that statistical thermodynamics models (most typically the Ising model) can indicate a preferred topology for network organization. Deco et al. used a straightforward Ising spin network to model brain Resting State Networks (RSNs) with coherent and robust spatiotemporal organization [86]. They found that system entropy increases markedly close to a bifurcation point, and that the RSN topology most closely matches a scale-free network topology (with hubs), as compared to other potential topologies, such as random, regular, or small-world. A key feature in their findings is that the scale-free topology (marked by central hubs) gives rise to a richer dynamic repertoire than is allowed with other topologies.

Deco et al. also found that the scale-free topology is associated with a specific small range of interaction parameters (*h*) for the Ising network with simple entropy formulation. They constructed networks with various topologies and then computed the entropy associated with these networks, in order to find those with both maximal entropy and topologic similarity to connectivity patterns evidenced in imaging studies. Their thesis is that increasing the entropy of the system (with more attractor states) allows for greater richness in the allowable dynamical repertoire. They hypothesize that high-entropic states are thus crucial to allowing the brain to have greater latitude for computation.

Their work introduces a potential connection to a more comprehensive formulation of the entropy, in that a scale-free or hub-based topology may potentially be characterized by systems with a given *h*-value, where the *h*-value directly indicates certain *configuration variables* associated with specific topographies. This is one of the next steps in an evolving research program.

Thus, the process of free energy minimization can play a role in network behaviors. A neural system comes to a base state of equilibrium. A stimulus causes the system to respond, and it enters into a state where certain patterns become active. It holds this state for a period of time, and potentially even learns to stabilize this as a new equilibrium state; this can happen only if different enthalpy parameters characterize this state vis-a-vis the prior resting state. A system can be perturbed, but it would come to a new (or learned) pattern–response–equilibrium state.

Friston, describing how such perturbations can influence a neural system, notes that “The emerging picture is that endogenous fluctuations are a consequence of dynamics on anatomical connectivity structures with particular scale-invariant and small-world characteristics” and that these models “... also speak to larger questions about how the brain maintains itself near phase-transitions (i.e., self-organized criticality and gain control)” [87].

Movement towards a new state will be influenced by the nature of the stimulus or perturbation. Friston describes this, saying: “In short, the internal states will appear to engage in Bayesian inference, effectively inferring the (external) causes of sensory states. Furthermore, the active states are complicit in this inference, sampling sensory states that maximize model evidence: in other words, selecting sensations that the system expects. This is active inference...” [88].

### 3.2. Statistical Thermodynamics for Phase Transitions and Critical Points in Neural Assemblies

Over the past three decades, various researchers have evolved an understanding that the brain self-organizes to operate at a critical or near-critical threshold. There is a relationship between network topography and the brain’s ability to exhibit spontaneous dynamics (“neural avalanches”), so that different “connectivity rules” drive the brain towards different dynamic states. Massobrio et al. (2015) argue for this understanding with support drawn from a combination of in vitro studies together with theoretical investigations [89].

Kozma, Puljik, and Freeman “treat cortices as dissipative thermodynamic systems that by homeostasis hold themselves near a critical level of activity that is far from equilibrium but steady state, a pseudo-equilibrium” [90]. They use the notation for K (Katchalsky) sets based on Freeman’s notions of a hierarchical brain model, and employ a neuropercolation approach to “[model] brain dynamics ... as a sequence of intermittent phase transitions in an open system”.

Deco et al. similarly advocate the notion that Resting State Networks (RSNs) operate near a critical point, noting that “at rest, the brain operates in a critical regime, characterized by attractor ghosts or weak attractors in the Milnor sense. This leads to chaotic itinerancy—which may be augmented by random fluctuations and an exploration of multistable attractors (in the strong or classical sense). When a stimulus is encountered (or attentional set changes), one attractor becomes stable and is ’selected’ from a repertoire of latent attractors” [86].

They follow this work with using a dynamic mean-field approximation for neural interactions, which approximates the temporal dynamics of the spiking network [91]. They were able to show that functional connectivity emerges from slow linear explorations in the region of system bifurcation. They suggest that small fluctuations around the spontaneous state better explain the resting state brain dynamics, compared with large excursions shaped by attractor basins. They comment that “... working at the edge of a critical point allows the system to rapidly compute a specific brain function by representing it in an attractor”. Freeman has summed this up by stating that “The cerebral cortex [achieves a critical state] by maintaining itself at the edge of chaos (Nicolis, 1987 [92,93]) in a state of criticality (Kozma et al., 2012 [90])” [94].

One implication for this, as we model brain states with a free energy minimization approach, is that small fluctuations around local pattern configurations can potentially be a means by which these “slow linear explorations” of the state space can occur.

There are, of course, opposing points of view on the nature of critical phenomena in the brain [95]. However, strong support for the notion that retinal circuits, at least, operate within the critical region comes from recent work by Kastner et al., who show that neural circuits operating near a critical point maximize information encoding [96]. With this connection between neural structural and functional architecture and network dynamics, it is reasonable to investigate how the local pattern distributions, as given by configuration variables, correspond with various cluster architectures (random, small-world, and scale-free [89]).

Massobrio et al., studying the self-organized criticality in cortical assemblies which occurs in concurrent scale-free and small-world networks, note that “The obtained results support the hypothesis that the emergence of critical states occurs in specific complex network topologies” [89].

Numerous other studies, most within the past few years, support the existence of critical points within close reach of ongoing neural dynamics. For example, Abum et al. find—by tuning background excitation to induce a Hopf bifurcation approach—that there is a dramatic increase in the autocorrelation length, depending sensitively on which neuronal subpopulation is stochastically perturbed [97].

There is also recent work suggesting that neural systems operate in the subcritical regime. Priesemann et al. find that—in contrast to the nominally suggested self-organizing critical (SOC) systems—neural avalanches observed in vivo imply that neural activity does not reflect a SOC state but a slightly sub-critical regime without a separation of time scales. They note that the potential advantages of operating in the subcritical regime may be faster information processing, together with a safety margin from super-criticality [98]. Tomen et al. find that emerging synchrony in the visual cortex of macaque area V4 yields an increase in selective discriminability. However, this only occurs within a narrow region in the phase space, at the transition from subcritical to supercritical dynamics [99].

Although there are a wide range of dynamic behaviors, the existence of long-term attractor states provides a stabilizing backdrop against which many other activities play out. Braun and Mattia describe various attractor states relevant to this work, including hierarchical nested attractors [100].

Recently, various researchers have introduced significant findings that strongly influence the design of a neuromorphic system that takes advantage of statistical thermodynamics principles. One example is that of Butler et al., introducing a key finding with implications for the design of neuromorphic systems as well as for the direct modeling of neural activation patterns. This finding is that in order to avoid destabilization (e.g., into hallucinogenic states), certain areas of the visual cortex require sparse long-range connections in addition to their local connectivity [101]. Butler et al. interpret this as an evolutionary adaptation, tuning the visual processes in the V1 cortex.

Cowan et al. (2013) examine a simple neural network that exhibits both self-organization and criticality, along with hysteresis. Using renormalization group analysis, they find that the network states undergo the neural analog of a phase transition in the universality class of directed percolation [102].

### 3.3. Criticality, Complexity, and Entropy: A Rationale for High-Entropic Systems

While the previous subsection focused on the role of criticality in brain systems, together with the role of entropy as the crucial mediating process, the notion of complexity is perhaps even more important for information processing within neural systems.

Nicolis introduces how biological systems differ from simpler mechanistic information storage and transmission systems by invoking the notion of complexity versus entropy [92,93,103]. He states regarding the notion of information that “It is rather emerging from the iterated map that the processor applies by way of conjugating externally impinging stimuli with internal activity...” [93]. In short, information is not the storage nor the time series itself, but rather the stable unfolding process; specifically, the “set of (stable) eigenfunctions of such a map—which for a dissipative nonlinear operator is simply the set of the coexisting attractors.”

In a different but related line of thought, Crutchfield and Young (1989) introduced an abstract notion of complexity, and devised a procedure for reconstructing computationally equivalent machines (or *ε*-machines) whose properties lead to quantitative estimates of both complexity and entropy [104]. In short, certain lines of research lead to the association of neuronal connectivity (both structural and functional) not only with critical phenomena, but also with potential means for describing complexity and higher-level descriptions of information communication within the brain.

The CVM approach (including 2-D versions) yields critical behavior, and it is thus reasonable to consider using this approach when modeling neural behavior [105]. Connections between the CVM and various topologies need to be investigated, and a potential further linkage to complexity measures would be a follow-on endeavor.

Following lines of thought introduced by Friston [83], from the perspective of the free energy principle (i.e., the minimization of variational free energy), both self-organized criticality and complexity reduction are mandatory. This follows because the variational free energy can always be expressed as complexity minus accuracy, when the enthalpy in turn is interpreted as a log probability density. This means that minimizing free energy necessarily entails a minimization of complexity. In this context, complexity is defined as the Kullback–Leibler (KL) divergence between posterior and prior distributions. In other words, complexity is the degree to which the distribution is pulled away from the prior (equilibrium) distribution upon exposure to the environment [13,106,107].

Interestingly, the variational free energy can also be expressed as entropy minus enthalpy. The entropy term (under some simplifying—Laplacian— assumptions) corresponds to the curvature of the free energy. This means that a minimum free energy solution, in certain situations, occupies a regime of the free energy that has low curvature. In turn, this means that local perturbations induce unstable critical or slow fluctuations. In other words, minimizing free energy necessarily leads to a critical slowing [13].

The immediately following Section 4 introduces the *configuration variables*, and shows visually and intuitively how different sets of local patterns correspond with different values of the interaction enthalpy parameter *h*.

The two succeeding sections introduce the theoretical formalism of the CVM; first for a 1-D (Section 5), and then a 2-D array (Section 6).

## 4. Configuration Variables: Describing Local Patterns

This section describes the *configuration variables* used in the ***cluster variation method (CVM)***; a method for modeling the free energy that takes into account not only distributions of units into **A** and **B** states, but also the distributions of local patterns given in terms of *configuration variables*.

The configuration variables score the frequency with which various local configurations would be sampled from a lattice. In other words, they are the fraction of times they occur under random sampling. This means that they can be interpreted in terms of the probability of finding a particular configuration of one or more states when sampling at random or over long periods of time (under ergodicity assumptions). We will appeal to their interpretation as probabilities later when linking the configuration variables to entropy.

### 4.1. Configuration Variables in the Context of Entropy

Before introducing the formalisms, it may be useful to appeal to our intuition about the nature of entropy. The following Figure 1 shows a set of three patterns, each arranged as a 1-D zigzag chain. If we use only our classic entropy expression, which depends only on the distribution of units into *on* and *off* (or **A** and **B**) states, then the entropy for each of these patterns would be the same. In fact, for this illustration, the entropy for each would be at a maximum, since there are equal numbers of **A** and **B** units in each case.

Appealing to our intuitive sense about the nature of entropy, we immediately recognize that the top and bottom patterns in this figure are substantially *not* at maximal entropy; each of these two patterns expresses a high degree of order (albeit with a small pattern discontinuity in each). Our understanding of the concept of entropy tells us that neither pattern A (topmost) nor pattern C (bottom-most) is achieving a maximal distribution among all available states. (Obviously, the notion of *states* here has to be broadened to include patterns as well as distribution into **A** and **B** states.)

It is the middle pattern (B) of Figure 1 that intrigues us the most at this moment. As with both the top (A) and bottom-most (C) patterns, it is not entirely a maximal distribution among possible states; we see that the same eight-unit pattern is repeated four times. However, despite this regularity, we notice that there is a greater diversity of local patterns in this middle example than in either the topmost or bottom-most examples.

The key to expressing this diversity among the types of local patterns, and to noting their resulting impact on the entropy term, is to consider the local *configuration variables*.

We have an analytic solution giving the configuration variables as functions of the interaction enthalpy *h* (details to follow in Section 5). This gives us a high-level interpretation for the kinds of patterns that we have within a dataset: are the nearest-neighbors typically like each other (ferromagnetic), or unlike (anti-ferromagnetic)? Using *h*, we know the probabilities of what kinds of unit will follow a given instance of a nearest-neighbor pair of units.

### 4.2. Introducing the Configuration Variables

This subsection sets the stage for the 1-D CVM method by:
Precisely articulating the *configuration variables*, andIdentifying their inter-dependencies.

A following subsection will show how the *h*-values given in Figure 1 (on the far right of each pattern) were obtained.

Following [6,7], we create a 1-D lattice as a single zigzag chain composed of two staggered sets of *M* units each as shown in Figure 2. A single zigzag chain is used here for two reasons. First, it lends itself readily to constructing and depicting the next-nearest-neighbor and triplet configurations, and was in fact part of the original CVM formulation [6,7]. Second (and perhaps more significantly for our purposes), it provides a means of depicting M=3 functional and structural units, so it will be feasible in future work to map abstract neurophysiological networks to a CVM representation.

Figure 2 illustrates the three configuration variables:
$x_{i}$ - Single units,$y_{i}$ - Nearest-neighbor pairs,$w_{i}$ - Next-nearest-neighbor pairs, and$z_{i}$ - Triplets.

Figure 3 illustrates the *triplet* configuration variables $z_{i}$ for a bistate system.

Notice that within this figure, the triplets $z_{2}$ and $z_{5}$ have two possible configurations each: **A**-**A**-**B** and **B**-**A**-**A** for $z_{2}$, and **B**-**B**-**A** and **A**-**B**-**B** for $z_{5}$.

The degeneracy factors $\beta_{i}$ and $\gamma_{i}$ (number of ways of constructing a given configuration variable) are shown in Figure 4; $\beta_{2}=2$, as $y_{2}$ and $w_{2}$ can be constructed as either **A**-**B** or as **B**-**A** for $y_{2}$, or as **B**- -**A** or as **A**- -**B** for $w_{2}$. Similarly, $\gamma_{2}=\gamma_{5}=2$ (for the triplets), as there are two ways each for constructing the triplets $z_{2}$ and $z_{5}$. All other degeneracy factors are set to 1.

The relations among the configuration variables are given in Appendix A.

### 4.3. A Glance at h: The Interaction Enthalpy Corresponds to Observed Configuration Values

In Figure 1, we observed three different illustrative sets of local patterns, each expressed as a 1-D zigzag chain.

Intuitively, we understood that each pattern had a different entropy, dependent on local pattern configurations:
A: Predominantly like-near-like (“ferromagnetic”), with two small exceptions,B: Diverse local patterns, representing the equilibrium distribution of patterns when the interaction enthalpy is set to zero, andC: Predominantly like-near-unlike (reading the nearest-neighbor as the *diagonal connection* (“anti-ferromagnetic”), again with two small exceptions.

This figure presented *h*-values associated with the three different patterns, A–C. Our intuitive sense of maximizing entropy at equilibrium leads us to think that pattern B would have the highest entropy of the three.

Now we are in a position to express our intuitive understanding in terms of the *configuration variables* and the values which they assume in each of these distinct patterns.

With formalizations for the various configuration variables ($x_{i}$, $y_{i}$, $w_{i}$, and $z_{i}$) defined in the previous subsection, as well as in in Appendix A, we can now determine the $z_{i}$ (the triplet fractional values) for each pattern (A–C). Additionally, we can now interpret how *h* (the interaction enthalpy parameter) corresponds to the $z_{i}$ (triplet distributions).

Note that the values for $h_{avg}$ (as shown in Figure 5) are computed as the mean of $h_{1}$ and $h_{3}$, which are each based on computing *h* corresponding to the values for $z_{1}$ and $z_{3}$, respectively. The *h* computations are done via a look-up table based on results given in Section 5.5, and specifically as shown in Figure 6. The values for $h_{1}$ and $h_{3}$ differ from each other slightly, because the example systems shown in Figure 1 and Figure 5 are of such limited scale that a perturbation of even a single unit moves the system away from equilibrium. This difference between values would disappear for systems with larger total numbers of units.

Referring to Figure 5 (and also Figure 1), we notice that pattern **A** is largely like-near-like (or ferromagnetic), and has an $h_{avg}$ value of 1.97. The bottom-most pattern **C** is largely composed of unlike pairings (reading nearest-neighbors as the units on the diagonal), and thus could be described as antiferromagnetic. It has an $h_{avg}$ of 0.454. The central pattern **B** has an equilibrium distribution of configuration variables, and an $h_{avg}$ of 1.

As calculations in Section 5 and Section 6 will show, the configuration variables are a function of the parameter *h*, and we can use a look-up table and extrapolation to obtain the approximate value for *h* given a certain set of configuration variables. For the moment, we use the computed *h*-values to gain an intuitive understanding of how the interaction enthalpy parameter *h* influences the equilibrium distribution of local patterns.

Thus, even before we proceed with the mathematical treatment, we note that $h_{avg}>1$ indicates that the overall free energy is minimized (a free energy equilibrium point is found) when nearest neighbors are like each other, and that $h_{avg}<1$ indicates that the free energy minimum is found when nearest neighbors are unalike. The central pattern B, where $h_{avg}=1$ corresponds to the case where the interaction enthalpy between nearest-neighbor pairs is zero, and thus the maximal distribution among possible local patterns is observed.

We thus have the beginning for characterizing systems in terms of their local pattern distributions using only a single parameter, *h*.

The following Section 5 provides the theoretical basis for what we have just observed.

## 5. The Cluster Variation Method for a 1-D Zigzag Chain

The cluster variation method (CVM) was introduced by Kikuchi in 1951 [6], refined by Kikuchi and Brush in 1967 [7], and further refined by Miyata et al. in 1975 [108]. Sanchez et al. have provided a generalized formalism for this method [109], and Mohri [110] has presented substantial theoretical developments, applying the CVM to free energy phase space diagrams for various alloys. The CVM is a means of considering the entropy of a system as being more than a simple distribution among the allowable states for individual units. Rather, it encompasses the local patterns, considering nearest-neighbor, next-nearest-neighbor, and other clusters.

Pelizzola [111] has reviewed the CVM and its many applications, including how it relates to graph theory, as has Yedidia et al., who have identified how the CVM relates to belief propagation [112].

Barton and Cocco used Selective Cluster Expansion—very similar to a CVM approach—to model neural activity in the retina and the prefrontal cortex [76]. They found that their results are closest to those found using a simple mean-field theory with strongest interaction parameters.

The Barton–Cocco result will prove interesting and useful, as we discuss how neural functional and structural organization (specifically, “rich-club” and similar network topographies) will likely correspond with specific *h*-values in Section 7.

### 5.1. The Classic Free Energy Formulation

A system reaches free energy equilibrium when the free energy is at a minimum. The free energy itself reflects interaction between two functions: the enthalpy and the entropy.

The free energy of a system is given as
(1)G=H−TS
where *G* is the Gibbs free energy, *H* is the enthalpy, *T* is the temperature, and *S* is the entropy. The Gibbs free energy is one in which the system is closed, with constant pressure. The enthalpy term is a combination of internal energy (U) and a pressure–volume term. Naturally, as we anticipate both constant pressure and volume in the brain, the term measuring pressure change goes to zero. For the remainder of this article, the Gibbs free energy will simply be referred to as the free energy.

#### 5.1.1. The Enthalpy Term

In the earliest work on the Cluster Variation Method (CVM), Kikuchi found the free energy for his system using an enthalpy term given as:
(2)$$H=N\varepsilon (-y_{1}+2y_{2}-y_{3})$$
where *N* is the total number of units in the system and *ε* is the interaction energy between two nearest-neighbor units.

The physical interpretation of the preceding equation is that the nearest-neighbor interaction between two like units ($y_{1}$ and $y_{3}$) is stabilizing, or has a negative coefficient, and an interaction between unlike units ($y_{2}$) is destabilizing, or positive.

For this work, it is useful to shift the interaction energy base so that the interaction enthalpy between like units is zero, and the interaction enthalpy between unlike units ($y_{2}$), *ε*, is constant. This allows us to rephrase the enthalpy equation as:
(3)$$H=2N\varepsilon y_{2}.$$

The free energy is then:
(4)$$G=H-TS=2N\varepsilon y_{2}-TS$$

#### 5.1.2. The Entropy Term

The classic entropy formulation, using a reduced form that will give a dimensionless expression for the entropy–temperature term, is given as
(5)$$\bar{S}=S/\left(k_{\beta }N\right)=-\sum x_{i}ln\left(x_{i}\right)$$
where $k_{\beta }$ is Boltzmann’s constant. Note that the set of variables $x_{i}$ in this equation refers to single units, and does not as yet specify that the units should occur in either of states **A** or **B**. Further, this equation does not express the influence of configuration variables, which will be addressed shortly.

For a typical bistate system, the above gives
(6)$$\bar{S}=-\left(x_{1}ln\left(x_{1}\right)+x_{2}ln\left(x_{2}\right)\right)$$

We also note, letting $x_{1}=x$ and $x_{2}=1-x$ (which is allowed, because $x_{1}+x_{2}=1$), that
(7)$$\bar{S}=-\left(xln\left(x\right)+\left(1-x\right)ln\left(1-x\right)\right)$$

We insert the entropy formulation into the previous free energy equation, and create an equation for $\bar{G}$ (the reduced free energy), where $\bar{G}=G/\left(Nk_{\beta }T\right)$; this allows us to work with dimensionless values. This yields
(8)$$\bar{G}=2\varepsilon y_{2}/\left(k_{\beta }T\right)+xln\left(x\right)+\left(1-x\right)ln\left(1-x\right)$$
where the term $k_{\beta }T$ within the first term on the right hand side of the equation represents a constant that can be set equal to one, as we are dealing with an abstract representation of free energy, and not one tied directly to temperature.

### 5.2. The Entropy for a 1-D Zigzag Chain

To more formally generate the entropy term, we write the entropy of the system as the natural logarithm of the Grand Partition Function Ω:
(9)$$S=k_{\beta }ln\Omega$$
where Ω, the degeneracy factor (Grand Partition Function) is the number of ways of constructing the system in such a way that the fraction variables take on certain values.

Following the approach introduced by Kikuchi [6,7], we begin by considering a one-dimensional system composed of a single zigzag chain, as was shown in Figure 2.

Details for constructing the Grand Partition Function from which we obtain the entropy term are presented in Appendix A. The resulting entropy expression for the 1-D CVM is:
(10)$$S_{1-D}=k_{\beta }ln\Omega_{double}=2k_{\beta }\left[\sum_{i=1}^{3}\beta_{i}Lf\left(y_{i}\right)-\sum_{i=1}^{6}\gamma_{i}Lf\left(z_{i}\right)\right],$$
where Lf(x)=xln(x)−x.

### 5.3. The Free Energy Equation for the 1-D Zigzag Chain

We define the reduced free energy for a 1-D system, ${\bar{G}}_{1-D}$, in terms of the free energy for a 1-D system as
(11)$${\bar{G}}_{1-D}=G_{1-D}/\left(Nk_{\beta }T\right),$$
where as before, the term $k_{\beta }T$ will be set equal to one. The total number of units in the 1-D system is N=2M, as the system is comprised of two staggered sets of length *M* each in order to create a single zigzag chain (see Section 4.2, and also Figure 2). We thus create a reduced equation, normalized with respect to both temperature and the total number of units. The temperature, along with Boltzmann’s constant, will be subsumed into the enthalpy parameter, creating a dimensionless term representing interaction strength between two nearest-neighbor units.

We introduce both the entropy for the 1-D chain as well as the pairwise enthalpy into the free energy equation, where the entropy term now not only represents the distribution of units into active/inactive states, but also the distribution of local patterns or configurations. We take the case where the enthalpy-per-active unit is zero, and thus only deal with the interaction enthalpy term.

We also make use of a substitution expressing the configuration variables in terms of each other (identified in Appendix A):
(12)$$\begin{array}{c} y_{2}=z_{2}+z_{4}=z_{3}+z_{5} \end{array}$$
(13)$$\begin{array}{c} 2y_{2}=z_{2}+z_{4}+z_{3}+z_{5} \end{array}$$

This results in a much more complex reduced equation than the standard Ising (Bethe–Peierls) formulation [113,114] (notation drawn from [115,116], see [117]), and where the term $k_{\beta }T$ has been set equal to one.
(14)$$\begin{array}{cc} {\bar{G}}_{1-D}=G_{1-D}/N= & \\ & \varepsilon \left(z_{2}+z_{3}+z_{4}+z_{5}\right)-2\sum_{i=1}^{3}\beta_{i}Lf\left(y_{i}\right))+2\sum_{i=1}^{6}\gamma_{i}Lf\left(z_{i}\right)\\ + & \mu \left(1-\sum_{i=1}^{6}\gamma_{i}z_{i}\right)+4\lambda \left(z_{3}+z_{5}-z_{2}-z_{4}\right) \end{array}$$
where *μ* and *λ* are Lagrange multipliers.

Taking the derivative of $\bar{G}$ with respect to the six configuration variables $z_{i}$, and setting each derivative equal to zero yields the following six equations, presented in detail in Appendix A:
(15)$$\begin{array}{ccc} z_{1}q & = & y_{1}\\ z_{2}q & = & {\left(y_{1}y_{2}\right)}^{1/2}e^{-\varepsilon /4}e^{\lambda }\\ z_{3}q & = & y_{2}e^{-\varepsilon /2}e^{-2\lambda }\\ z_{4}q & = & y_{2}e^{-\varepsilon /2}e^{2\lambda }\\ z_{5}q & = & {\left(y_{2}y_{3}\right)}^{1/2}e^{-\varepsilon /4}e^{-\lambda }\\ z_{6}q & = & y_{3} \end{array}$$
where $q=e^{-\mu /2}$, and *μ* can be shown to be (for chemical systems) the chemical potential.

### 5.4. Analytic Solution for the CVM for the 1-D Zigzag Chain

Achieving the analytic solution involves solving a free energy equation for the equilibrium point, where the enthalpy-per-active-unit is given as zero, and the pairwise-interaction enthalpy is given as *ε*. Because the enthalpy-per-active-unit is zero, there is no a priori preference for a unit to be in either the **A** or **B** states; thus, $x_{1}=x_{2}=0.5$.

This gives a symmetric distribution to the other configuration variables; specifically:
(16)$$\begin{array}{ccc} y_{1} & = & y_{3}\\ z_{1} & = & z_{6}\\ z_{2} & = & z_{5}\\ z_{3} & = & z_{4} \end{array}$$

The resulting solution gives the equilibrium configuration variables $y_{i}$ and $z_{i}$ in terms of *h*, where h=exp(ε/4).

For the system where $x_{1}=x_{2}=0.5$ and λ=0, we can solve the previous equations, giving $z_{i}q$ in terms of the $y_{i}$ and other parameters to obtain the fraction variables $y_{i}$ and $z_{i}$. The calculations, briefly summarized in the following paragraphs, are presented in more detail in Appendix A.

Let $h=e^{\varepsilon /4}$, and $s=z_{1}/z_{3}$. Then, as a step in the derivation, we obtain
(17)$$h^{2}=s\frac{1+s^{1/2}}{s+s^{1/2}}$$
or
(18)$$s=h^{4}$$

The result is:
(19)$$1/z_{3}=2\left(s+1\right)\pm 4s^{1/2}$$

A slightly expanded summary is found in in Appendix A. A fully-detailed derivation is found in [9].

### 5.5. Results for the 1-D CVM

In order to introduce the overall CVM approach, this subsection continues to focus on the 1-D CVM as the simpler and more intuitive result. The following Section 6 introduces the 2-D CVM, for which the equations are somewhat more complex. However, similar lines of thought will apply.

This subsection shows how the configuration variables $y_{i}$ and $z_{i}$ are mapped to a single *h*-value at equilibrium. It also reveals how to interpret the configuration variables as functions of *h*.

The analytic solution for the 1-D CVM starts by giving one of the configuration variables ($z_{3}$) as a function of the reduced interaction energy term *h*. From this, the remaining configuration variables are found as functions of *h*. The full details of this work were presented in Maren [9].

A similar analytic solution for the 2-D CVM has been presented in Maren [10]. Both analytic results are based on original derivations in Maren [85].

Figure 6 shows three of the configuration variables, $z_{1}$, $z_{3}$, and $y_{2}$, in terms of the parameter *h*.

The following three subsections assist in interpreting the results shown in Figure 6; these are the cases where the interaction enthalpy:
ε=0 so that $h=e^{\varepsilon /4}=1$,ε<0 so that $h=e^{\varepsilon /4}<1$, andε>0 so that $h=e^{\varepsilon /4}>1$.

### 5.6. Configuration Variables When the Interaction Enthalpy ε=0 (h=1)

We begin by noting (in the slightly left-of-center portion of Figure 6 where h=1) that the observed configuration variable values are exactly as we would expect. Specifically, $z_{1}=z_{3}=0.125$, and $y_{2}=0.250$. This fits precisely with our expectation for the configuration variables at equilibrium when ε=0 ($h=e^{\varepsilon /4}=1$).

To understand intuitively that this would be the expected result, we reflect on the total number of triplet states available, together with their degeneracies. We use various normalization equations from Appendix A as well [9], and particularly note that
$$1=x_{1}+x_{2}=\sum_{i=1}^{6}\gamma_{i}z_{i}$$

Since we also have $x_{1}=x_{2}=0.5$, as well as the symmetry results from the end of the last section, we have
$$0.5=y_{1}+y_{2}=y_{2}+y_{3}=\sum_{i=1}^{3}\gamma_{i}z_{i}$$

We also have that the degeneracy factors are $\beta_{2}=\gamma_{2}=\gamma_{5}=2$, with all other $\beta_{i}$ and $\gamma_{i}$ set to 1. At equilibrium with ε=0, we expect equiprobable distribution of the $z_{i}$ among all possible states, so that we expect $z_{1}=z_{3}=0.125$, as observed in Figure 6. Additionally (not shown in the Figure), we would expect that $z_{2}=0.250$, so that $z_{1}+z_{2}+z_{3}=0.5$.

### 5.7. Configuration Variables When ε<0 (h<1)

At the left-hand-side of the preceding Figure 6, we have the case where $h=e^{\varepsilon /4}<1$. These small values for *h* means that *ε*, the interaction enthalpy between two unlike units (**A**-**B**), is negative. This means that we stabilize the system by providing a structure that emphasizes alternate units (e.g., **A**-**B**-**A**-**B**), as was shown in Pattern C of Figure 1. (Recall that the nearest-neighbor connections are read as the diagonal nearest-neighbors in this zigzag chain.)

This is precisely what we observe. The pairwise combination $y_{2}$ (**A**-**B**) increases beyond the nominal expectation (when there is no interaction energy), so that $y_{2}\to 0.5$, notably when h<0.1.

As a natural consequence of the increase in $y_{2}$ when h→0, we also have $z_{3}\to 0.5$ (maximizing **A**-**B**-**A** triplets) and also $z_{1}\to 0.0$ (minimizing **A**-**A**-**A** triplets).

### 5.8. Configuration Variables When ε>0 (h>1)

Consider the case of a positive interaction energy between unlike units (the **A**-**B** pairwise combination). This is the case on the right-hand side of Figure 6, where ε>0 then yields $h=e^{\varepsilon /4}>1$.

The positive interaction energy (ε>0) suggests that a preponderance of **A**-**B** pairs ($y_{2}$) would destabilize the system, as each **A**-**B** pair would introduce additional enthalpy to the overall free energy, which is at a minimum in the equilibrium state. We would thus expect that as *ε* increases as a positive value, we would see a decrease in $y_{2}$, and also see smaller values for those triplets that involve non-similar pair combinations. That is, we would expect that the **A**-**B**-**A** triplet, or $z_{3}$, would approach zero.

This is exactly what we observe this on the right-hand side of the Figure 6.

Thus, when h>>1 (or ε>>0), we see that $z_{3}$ falls towards zero, and $y_{2}$ decreases as well. Note that $y_{2}$ can never go towards zero, because there will always be some **A**-**B** pairs in a system that contains units in a mixture of states **A** and **B**. Correspondingly, this is also the situation in which $z_{1}=z_{6}$ becomes large; for example, we see that $z_{1}>0.4$ when h>3.0. Notice that $z_{1}$ cannot approach 0.5, as $z_{3}$ did for the case where h<<1. The reason is that $z_{1}$ represents **A**-**A**-**A** triplets, and we will always have some **A**-**A**-**B** and **B**-**A**-**A** triplets, because we have equiprobable distribution of units into states **A** and **B**. Thus, while we could conceivably have constructed a system composed exclusively of **A**-**B**-**A** and **B**-**A**-**B** triplets, we cannot do so exclusively with **A**-**A**-**A** and **B**-**B**-**B** triplets.

The realm of h>>1 is one in which we observe a highly structured system where large “domains” of like units mass together. These large domains (each comprised of either overlapping **A**-**A**-**A** or **B**-**B**-**B** triplets) stagger against each other, with relatively few “islands” of unlike units (e.g., the **A**-**B**-**A** and **B**-**A**-**B** triplets). This is the case observed in Pattern A of Figure 1.

Naturally, this approach–using a “reduced energy term”, where we had set $\bar{G}=G/\left(Nk_{\beta }T\right)$ and the interaction enthalpy term was similarly reduced to $\varepsilon /(k_{\beta }T)$, does not tell us whether we are simply increasing the interaction energy or reducing the temperature; they amount to the same thing. Both give the same resulting value for *h*, and it is the effect of *h* that interests us when we map the CVM variables and (ultimately) the CVM phase space. Thus, because we are working in an abstract space, we have set $k_{\beta }T=1$ throughout the various steps. However, the role of temperature has been subsumed into the interaction enthalpy. Kastner et al. [96] identify that “The average noise across the neural population plays the role of temperature in the classic theory of phase transitions”.

## 6. The Cluster Variation Method for a 2-D Zigzag Chain

The preceding section served as a brief introduction and tutorial on how the interaction enthalpy parameter *h* influenced the distribution of the configuration variables. For ease of visualization, we examined the configuration variables in the context of a 1-D system, or a single zigzag chain.

For the purposes of modeling neural systems, we are far more interested in a 2-D system. This section presents the results for an analytic solution of the free energy minimum for the 2-D CVM when there is an equiprobable distribution of units into the **A** and **B** states.

### 6.1. Modeling Activation Patterns in 2-D Neural Assemblies

There is ample motivation for modeling patterns in 2-D systems:
Many neural patterns of activity are best modeled as dynamics in a 2-D system [118],Information is encoded as time-varying patterns in such 2-D arrays,The common approach to modeling information content addresses only distribution across two states (“on” and “off”, or **A** and **B**), not as patterns; a more comprehensive pattern characterization may prove to be a useful correlation to information measures.

We have known for some time that the brain makes nearly ubiquitous use of 2-D maps for information encoding. What we are understanding more recently is exactly how different types of neural connectivity operate within the 2-D maps to provide certain kinds of functional effectiveness. For example, Harris and Shepherd have described different neocortical circuit classes, including local and long-range connectivity along with other characteristics [119]. By characterizing these circuit classes, which are widespread in their appearance across different cortical areas, we have a more abstract description for local cortical organization. It may be possible to associate different cortical classes with specific local configuration variable distributions.

We also note that 2-D topographic maps are crucial not only for single-sensor modality representation, but also for sensor fusion, as initially identified in ground-breaking work by Stein and Meredith [120], and more recently comprehensively overviewed by Stein [121]. Thus, we need to model the statistical thermodynamics associated with pattern formation in 2-D systems.

### 6.2. Configuration Variables for the 2-D CVM

To introduce the 2-D CVM, we first examine a short 1-D CVM configuration, as shown in Figure 7. This figure illustrates (in an abbreviated sense) the same kind of pattern that we used as pattern B in Figure 1 and Figure 5. This figure shows us a pattern in which each of the $z_{i}$ triplets appears in their equilibrium values for the case where h=0. In particular, note that the total number of triplets (as countable numbers, $Z_{i}$, not just as the fractional values) is equal to the total number of units present. This is true only in the case where we have a horizontal wrap-around, so that there is no discontinuity at the borders.

We compare this with a similarly-patterned 2-D CVM system, as shown in the following Figure 8. In this figure, we have built up the 2-D overall configuration by adding two more single rows on top of the original single zigzag chain, creating a set of units that is double the original size. In order to maintain border continuity as we count the various configuration variables, we do an envelope-style wrapping vertically as well as horizontally.

It is worth noting here that the total number of countable triplets $Z_{i}$ is now *twice* the number of units in the system. Essentially, we can now count connections in both an up-and-down sense as well as sideways (all expressed on the diagonal, of course), which doubles the number of countable triplets.

With the previous Figure 8 as a guide to interpreting 2-D CVM patterns, we now examine a 2-D extension of the patterns offered earlier in the 1-D CVM. Previous Figure 1 and Figure 5 showed how the average *h*-values were associated with three different types of patterns (A–C). We now examine a similar (albeit 2-D instead of 1-D) representation in Figure 9.

Once again, we used the actual $z_{1}$ and $z_{3}$ values for each pattern to identify corresponding *h*-values, as we did earlier in the 1-D CVM (the following subsection gives highlights of the derivation for expressing configuration variables in terms of *h* for the 2-D CVM).

Just as we found in the case of the 1-D CVM, a higher value of $h_{avg}$ corresponds to the “like-near-like” pattern, whereas the lower value for $h_{avg}$ corresponds to the “like-near-unlike” pattern. Additionally, as before, when h=1, there is no interaction enthalpy between neighboring units, and thus the system falls into an equiprobable distribution of local configurations.

### 6.3. Free Energy in a 2-D Configuration

The equation for the free energy in a 2-D system, including configuration variables in the entropy term, is
(20)$$\begin{array}{cc} {\bar{G}}_{2-D}=G_{2-D}/N= & \\ & \varepsilon (z_{2}+z_{3}+z_{4}+z_{5})\\ - & \left[2\sum_{i=1}^{3}\beta_{i}Lf\left(y_{i}\right))+\sum_{i=1}^{3}\beta_{i}Lf\left(w_{i}\right))-\sum_{i=1}^{2}\beta_{i}Lf\left(x_{i}\right))-2\sum_{i=1}^{6}\gamma_{i}Lf\left(z_{i}\right)\right]\\ + & \mu \left(1-\sum_{i=1}^{6}\gamma_{i}z_{i}\right)+4\lambda \left(z_{3}+z_{5}-z_{2}-z_{4}\right) \end{array}$$
where (as before) *μ* and *λ* are Lagrange multipliers, and (as we did in the previous Section) we have set $k_{\beta }T=1$ (note: the full derivation of the 2-D CVM free energy is presented in [10], and the preceding equation corresponds to Equations (2)–(14) of that reference).

We note that in comparison with Equation (Equation 14), the equation for the 2-D CVM system (Equation (Equation 20)) includes terms in $w_{i}$ (next-nearest-neighbors) and $x_{i}$ (simple distribution into states **A** and **B**, which are set to be equiprobable for obtaining the analytic solution for minimum free energy).

Further, the enthalpy term in the preceding equation really should have a factor of two in it; that is, $2\varepsilon (z_{2}+z_{3}+z_{4}+z_{5})$. This is because when we move from a 1-D to a 2-D CVM, we double the number of configuration variables such as $y_{i}$ and $z_{i}$, as can be seen via visual examination of Figure 8 in comparison with Figure 7.

However, the values for parameter *h* itself are those that are the best means for comparing behaviors of the 1-D versus 2-D CVM equations, and it is more useful to simply subsume the factor of two into the term 2ε. As we will typically work within a 2-D CVM framework, the 1-D results are useful more as an introductory pathway than as an actual comparison of system dynamics. Our attention will ultimately be on those values of *h* (in the 2-D CVM) that give us system configurations that bear resemblance to interesting network topologies, such as small world, rich-club, etc.

Following methods very similar to those used to find the analytic solution to the 1-D CVM free energy at equilibrium, and again for the specific case where $x_{1}=x_{2}=0.5$, we obtain an analytic solution for the configuration variables.

The solution for the specific configuration variable $z_{3}$ is given as
(21)$$z_{3}=\frac{\left(h^{2}-3\right)\left(h^{2}+1\right)}{8[h^{4}-6h^{2}+1]}$$
where *h* has the same interpretation as in the previous section; that is, $h=e^{\varepsilon /4}$. Note that the preceding equation corresponds to Equations (2)–(17) of [10].

### 6.4. Results for the 2-D CVM

Figure 10 shows the results for the triplet configuration variable $z_{3}$ for both the 1-D and 2-D CVM at equilibrium (for the case where distributions into states **A** and **B** are equiprobable).

The analytic solution for the configuration variables in the 2-D CVM has divergence at two points, due to the presence of a quadratic term in the denominator. However, at values within a reasonable range of h=1, we have properties similar to those of the 1-D CVM configuration variables. Most specifically, we obtain the desired values at exactly h=1, and the values for the configuration variables trend as would be expected (until divergence causes less-than-welcome behaviors).

The following Figure 11 shows the values for $z_{3}$ in the neighborhood of h=1, for both the 1-D and 2-D CVM cases.

## 7. Discussion

This section briefly highlights important gleanings from the previous results, and overviews an intended research program, examining how a 2-D CVM can be used, and including two aspects:
Neural interactions giving rise to specific and changing *ε* values, andImplications for information content in patterns of neural activation.

One of the most interesting findings to emerge in studies of neural architecture is that “rich club” topographies play a strong role, as discussed previously in Section 2.2. As described in Nigam et al. (2016), “Neurons with the highest outgoing and incoming information transfer were more strongly connected to each other than by chance, thus forming a ’rich club’” [122].

When the rich node coefficient for a network is close to one, it signifies that high degree nodes of the network are well connected with each other. This means that the nodes which have greatest connectivity are mostly connected to other nodes which similarly have high connectivity, thus forming a densely-connected set of nodes, or a “rich club.”

Let us say that the nodes in state **A** are those that participate in a “rich club.” We would then expect those nodes to be strongly connected with each other, and not well-connected with nodes that are not in that state. If we were to suggest that proximal nearest-neighbor placement on a 2-D lattice would be conducive to forming such within-club connections, we would anticipate that we would have a network that largely has a “ferromagnetic-like” character; that is, like-near-like.

We would thus expect a topography in which $z_{1}$ (**A**-**A**-**A** triplets) is relatively high, and $z_{3}$ (**A**-**B**-**A**) is relatively small (see Figure 9). Thus, referencing Figure 10, we would expect to have *h*-values that are on the right-hand side of the graph; i.e., h>1, or equivalently, ε>0.

To be a bit more specific in our expectations, we refer again to Figure 9, and note the *h*-values corresponding to patterns A and B. For pattern B—which has a completely equiprobable distribution of triplets—h=1. For pattern A —which has two large masses of self-similar units, with just a few singleton-islands of opposing units within the larger masses—the *h*-value (average) is 2.42. Clearly, the kind of *h*-values that will govern brain networks will be somewhere between these two extremes.

As evidenced by the various works cited in prior sections, there is ample evidence that neural units in the brain organize into clusters that behave in a coherent manner. Thus, we do not expect anything like a completely random or equiprobable distribution of units. In short, we do not expect that *h*-values characterizing neural topographies will be near one.

On the other hand, something like pattern A from Figure 9 provides much too little in the way of diversity. If we mass all of our coherently-acting units together into a single, solid mass, then we lose the value provided by having different combinations of active units. Thus, *h*-values in the range of two or higher are also not at all likely to characterize neural topographies and activation patterns.

Having thus framed our expectations, the next round of work should yield more particular associations between *h*-values and network topographies, which should then correlate with our understanding of pairwise connection strengths. This would also be in keeping with results achieved by Barton and Cocco [76], and it also suggests an immediate task-list for investigation:
Identify *h*-value correlations with specific, characterizable rich-club, scale-free, and small-world topographies (to the extent that such can be expressed in a 2-D CVM lattice); note that this will likely involve configurations where $x_{1}\neq x_{2}$, and thus, computational as well as analytic solutions will be needed,Identify the actual free energies associated with those *h*-values and their related topographies, and ensure that any empirical distribution is indeed a minimum of a free energy functional, andCharacterize the shape of the free energy landscape in the vicinity of the various topographies, and in particular identify susceptibility to perturbations.

These tasks, particularly that of ensuring that the system is at a free energy minimum, will be essential prior to introducing perturbations and other steps that would invoke a variational principle of free energy minimization. This is simply a starter-set of tasks that will help us understand how the 2-D CVM can be useful in modeling neural structural and functional networks.

In addition to modeling neural systems directly, the CVM has potential as the internal or “hidden” layer in a neural network computational engine. One particular advantage of using the 2-D CVM in this role is that it is a natural means of implementing ideas suggested by Friston [82,83,88] requiring a computational engine whose free energy can be perturbed by input stimulus, and which can either return to its original state or find a new equilibrium state in response to perturbations.

This approach has, in fact, already been introduced [123,124,125]; the goal is now to move from the earlier instances of a 1-D computational engine to a 2-D, and to study performance under various conditions.

Once the basic performance of a 2-D CVM network as part of a computational engine has been understood, it will be possible to examine how the local structures expressed in such a network and characterized in terms of their enthalpy parameters can lead to a greater understanding of the information qualities of such configurations.

Over the long term, two specific lines of work will be to investigate the case of non-equiprobable distributions of units into states **A** and **B**; that is, where $x_{1}\neq x_{2}$. This will involve numerical calculations, as the analytic solution holds only for the case where $x_{1}=x_{2}$. However, this will yield rich information that can be applied to the neuronal modeling of various populations of active neural configurations. A second endeavor will be to perform kinetics calculations, using the Path Probability Method (PPM) as was devised by Kikuchi as the natural extension of the CVM to the time domain. The PPM will be a useful means of modeling how a system perturbed by a non-equilibrium process can move towards the final equilibrium state.

As a related theme, Feldman and Crutchfield have introduced information-theoretic measures of spatial structure and pattern in two dimensions, using the notion of excess entropy as a measure of complexity [126]. When the basic CVM network is more understood as a neural computational model, we will then be able to investigate complexity and other measures associated with such a 2-D system.

In relation to the above applications, if it can be shown that empirical neural network configurations minimize free energy under a given interaction enthalpy, then free energy becomes a Lyapunov function or Lagrangian for neuronal systems. This may have profound implications for elaborating variational principles of neuronal dynamics; e.g., gauge fields [106].

## 8. Conclusions

This paper has presented the 2-D cluster variation method from statistical thermodynamics, invoking the notion of entropy describing distributions among local pattern configurations as well as distributions among two possible states as a potentially useful means of characterizing neural systems.

Previous research has shown that nearest-neighbor interactions are significant in modeling neural systems, and the formulation proposed here relies exclusively on a nearest-neighbor interaction enthalpy. For the case where the distribution into states **A** and **B** is equiprobable ($x_{1}=x_{2}$), an analytic solution of the equilibrium values for configuration variables is possible, where the configuration variables can be expressed in terms of the interaction enthalpy parameter *h*.

The broader descriptive power of the 2-D CVM approach provides a means for studying neural network behaviors, and for modeling the information-processing capabilities of these networks. This is crucially important for next-generation Brain–Computer *Information* Interfaces (BCIIs), which will likely invoke densely-configured intracortical electrodes for much higher granularity in neural stimulus and signal recording.

The direction of next-generation Brain–Computer Interfaces will lead us into realms where the mapping of the external system to the neural substrate will not necessarily mimic existing body conditions [127]. One example of such is that of EEG-controlled robot swarms [128].

This will have huge repercussions for human self-perception. As Lakoff and Johnson described in their classic work, *Metaphors We Live By* [129], humans map a great deal of their experience into a somato-sensory reference frame. When we begin to integrate the perception of external systems that do not conform to our body-perception of self, we radically impact our notion of “what is self”. The societal implications will be enormous, ranging from experiential to philosophic, and from theological to legal.

It is into this emerging realm that we envision a closer coupling of modeling neural systems with computational platforms that can interpret neural activity. This will facilitate Brain–Computer Information Interfaces, and be part of an ever-evolving understanding of human experience and self-definition. Statistical thermodynamics has been shown to play a role in modeling neural systems. In the approach afforded by the CVM, we have an opportunity to extend the role of statistical thermodynamics in describing the brain, and thus link our understanding of a wide range of neural behaviors with a useful computational model.

## Figures and Tables

**Figure 1 brainsci-06-00044-f001:**
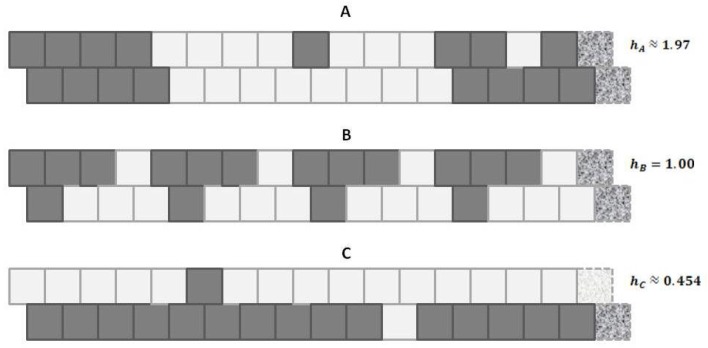
Three 1-D zigzag chains of 32 units each. The far right mottled units indicate wrap-around neighboring units from the far left; that is, the two units at the far right of each pattern (on the top and bottom rows) represent the two units at the farthest left; they are shown on the far right in order to make visualization of the nearest-neighbor, next-nearest-neighbor, and triplet clusters easier. In patterns A and B, the two units on the far right are mottled dark grey, corresponding to the two leftmost units that are shown in dark grey. In pattern C, the uppermost unit on the far right is shown in mottled light grey, corresponding to the light grey uppermost unit on the far left of pattern C. The dark grey mottling of the far-right unit on the bottom row of pattern C corresponds to the dark grey unit on the far left bottom row. Each pattern (A–C) has equiprobable *on* (state **A**) and *off* (state **B**) units. The three patterns can be described as: (A) ferromagnetic-like, with one inserted non-like unit in each domain; (B) a set of four eight-unit patterns with an equilibrium distribution of configuration triplets (achieved when the nearest-neighbor interaction energy is 0); and (C) anti-ferromagnetic-like, with one inserted like-near-like (ferromagnetic) unit of each type. In each case, the nearest-neighbor connection is read as the diagonal between two units. The parameter *h* (far right) indicates the nature and strength of pairwise interactions. Computation of the *h* parameter values for the above figure (and related discussion) is given in the following Section 5. In particular, the *h*-values are explained in detail in Figure 5 and in the corresponding text.

**Figure 2 brainsci-06-00044-f002:**
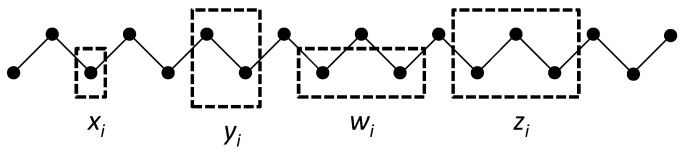
The 1-D single zigzag chain is created by arranging two staggered sets of *M* units each. The configuration variables shown are $x_{i}$ (single units), $y_{i}$ (nearest-neighbors), $w_{i}$ (next-nearest-neighbors), and $z_{i}$ (triplets).

**Figure 3 brainsci-06-00044-f003:**
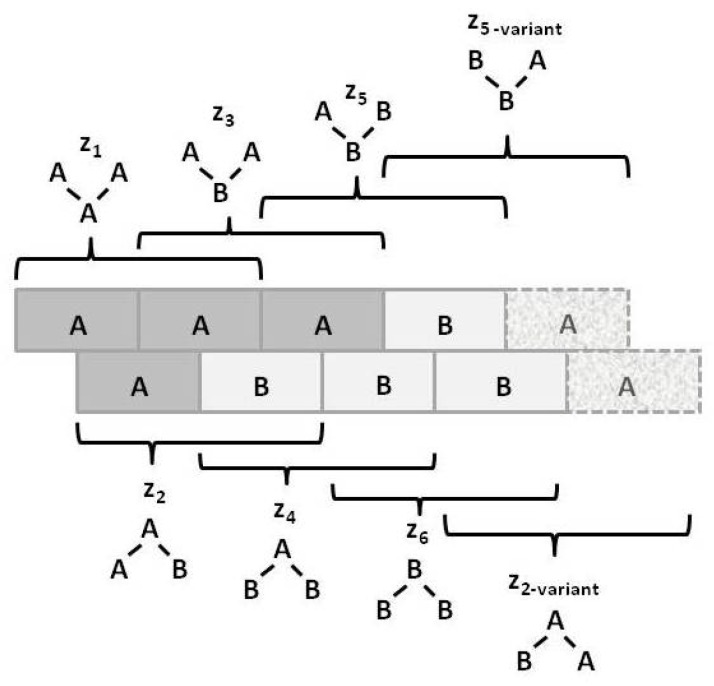
The six ways in which the configurations $z_{i}$ can be constructed.

**Figure 4 brainsci-06-00044-f004:**
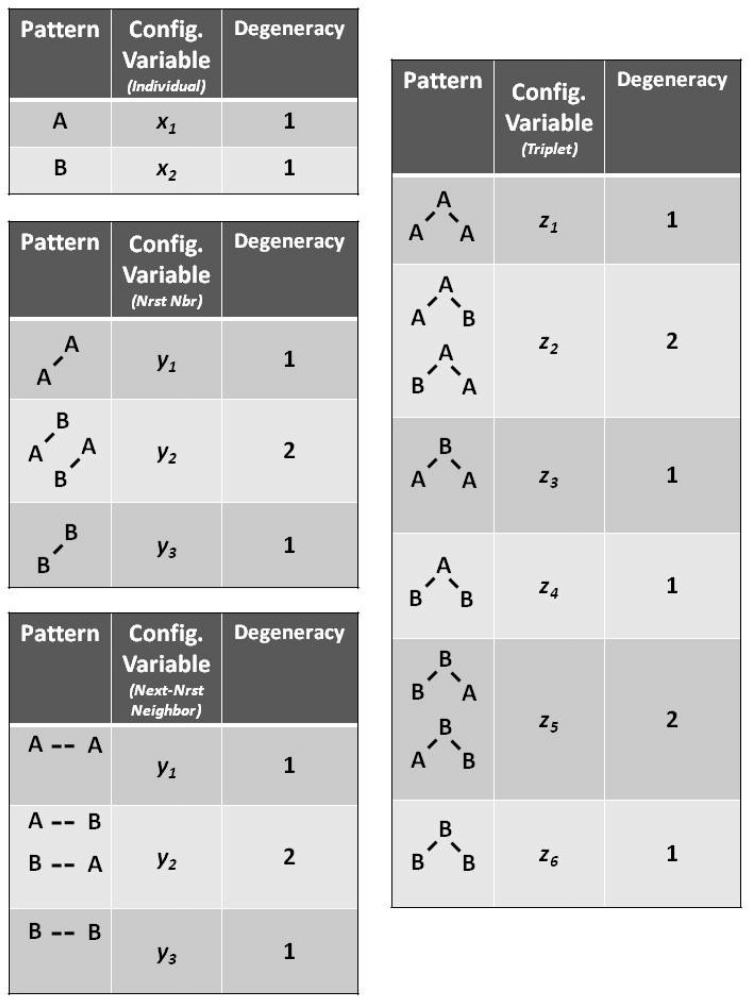
The ways in which the configurations variables $y_{i}$, $w_{i}$, and $z_{i}$ can be constructed, together with their degeneracy factors $\beta_{i}$ and $\gamma_{i}$.

**Figure 5 brainsci-06-00044-f005:**
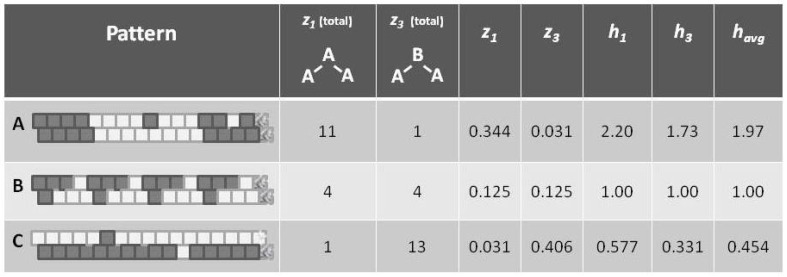
The previously-shown three patterns (A–C), as shown in Figure 1, can be characterized by their total values for $Z_{1}$ and $Z_{3}$ and their resulting fractional values, $z_{1}$ and $z_{3}$. Specific values for *h* correspond to each $z_{i}$ value. The resulting $h_{1}$ and $h_{3}$ values for a given pattern may differ, as they correspond to $z_{i}$ values for systems that are not at equilibrium. The $h_{i}$ values are averaged to produce $h_{avg}$.

**Figure 6 brainsci-06-00044-f006:**
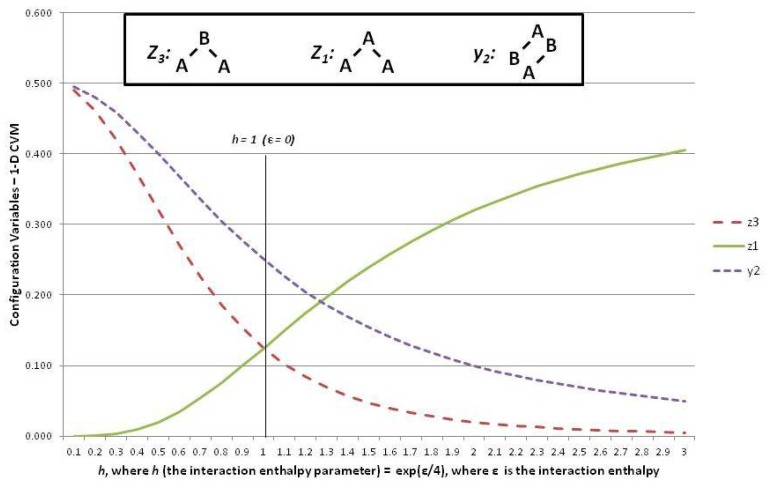
Results for three of the configuration variables, $z_{3}$, $z_{1}$, and $y_{2}$, used in the cluster variation method. Values for *h* are plotted along the x-axis.

**Figure 7 brainsci-06-00044-f007:**
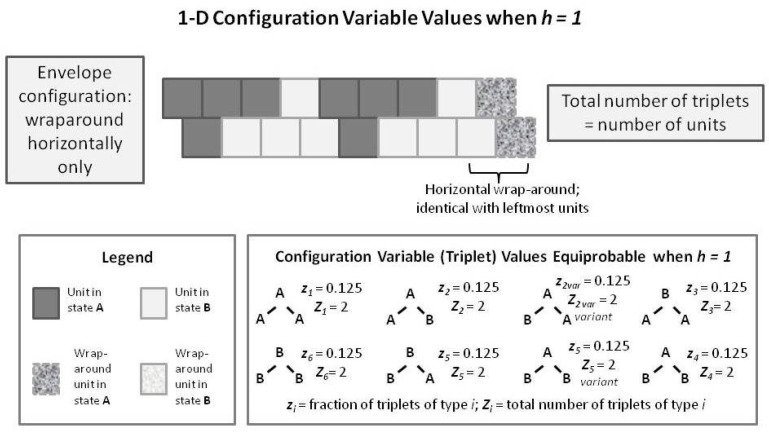
A 1-D CVM configuration (single zigzag chain), together with both the total numbers and fractional amounts for the configuration variables $z_{i}$. Notice that there is a horizontal wraparound for the computation of the configuration variables. The total number of triplets is equal to the total number of units in the system.

**Figure 8 brainsci-06-00044-f008:**
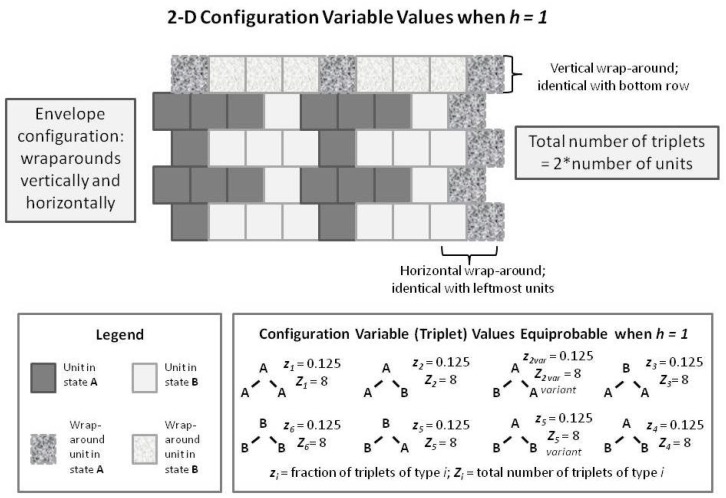
A 2-D CVM configuration (built by superimposing two single zigzag chains), together with both the total numbers and fractional amounts for the configuration variables $z_{i}$. Note that the wraparound in this case forms a full envelope; the configuration variables are computed in a smooth manner, wrapping both horizontally and vertically. The total number of triplets is equal to two times the total number of units in the system.

**Figure 9 brainsci-06-00044-f009:**
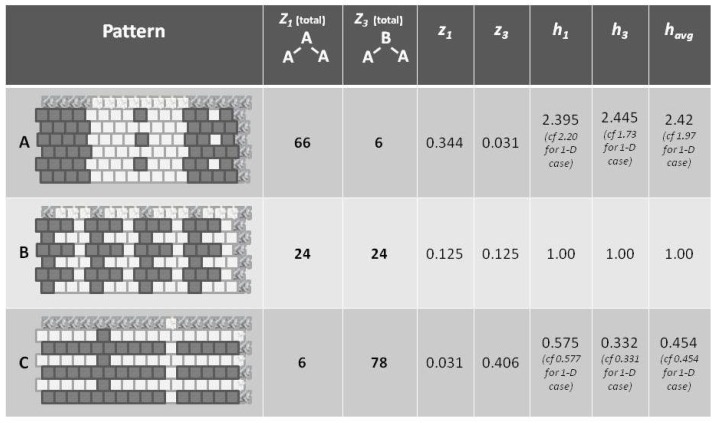
Replicating the kind of patterns shown previously for the 1-D CVM, the same patterns are extended to the 2-D CVM system. The actual fractional values for the various triplets are the same as for those used in the previous 1-D CVM figures.

**Figure 10 brainsci-06-00044-f010:**
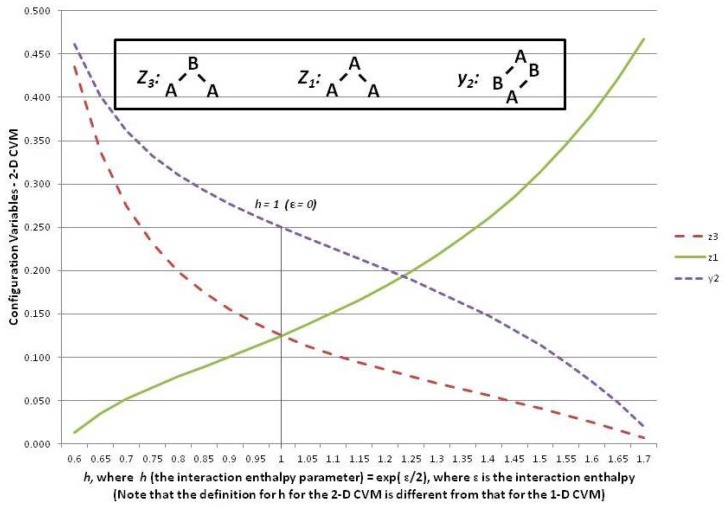
Comparison of configuration variable $z_{3}$ for the 1-D and 2-D CVM. Both values for $z_{3}$ are the expected 0.125 when *h* = 1.

**Figure 11 brainsci-06-00044-f011:**
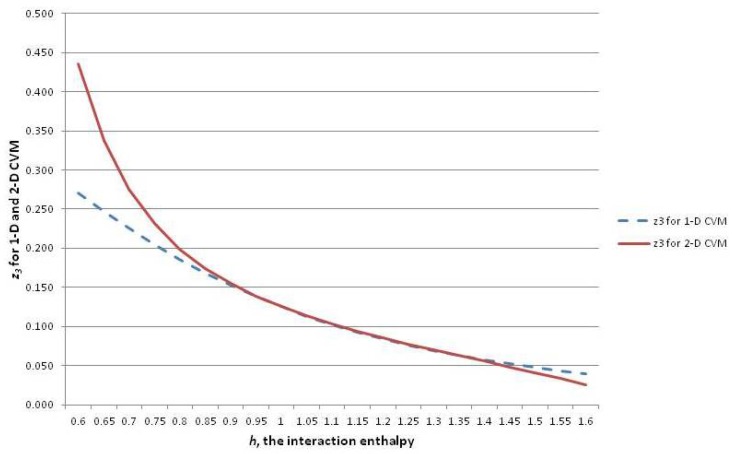
Comparison of configuration variable $z_{3}$ for the 1-D and 2-D CVM. Both values for $z_{3}$ are the expected 0.125 when *h* = 1.

**Table 1 brainsci-06-00044-t001:** Glossary of terms.

Term	Meaning
Configuration variable(s)	Nearest neighbor, next-nearest neighbor, and triplet patterns
Degeneracy	Number of ways in which a configuration variable can appear
Free energy	The thermodynamic state function *G*; where *G = H-TS*
Enthalpy	Internal energy *H* results from both per unit and pairwise interactions
Entropy	The entropy *S* is the distribution over all possible states
Equilibrium point	By definition, the free energy minimum for a closed system
Equilibrium distribution	Configuration variable values when free energy minimized for given *h*
Ergodic distribution	Achieved when a system is allowed to evolve over a long period of time
Interaction enthalpy	Between two unlike units, *ε*; influences configuration variables
Interaction enthalpy parameter	Parameter *h* is defined as $h=e^{\varepsilon /4}$
Motif	Fundamental building block; size (M) is equal to the number of vertices
Temperature	Temperature *T* times Boltzmann’s constant $k_{\beta }$ is set equal to one

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
