# Peer review of "The Cluster Variation Method: A Primer for Neuroscientists"

_brainsci, 2016, doi:10.3390/brainsci6040044_

Round 1

Reviewer 1 Report

The author applies Cluster Variation Method of statistical mechanics to investigate local pattern configurations and distributions between two possible states in neural systems.   Cluster Variation Method (CVM) was devised by late R. Kikuchi as a statistical mechanics method to deal with interacting particles, and later with the development of powerful computers CVM has been applied to various practical problems of Materials Science and Physics including theoretical calculations of phase diagrams, equilibrium configuration of atoms under a given temperature and interaction energies.  The author of this manuscript attempts to extend CVM to Brain-Computer Interfaces. The author’s knowledge of CVM is quite sufficient enough to bridge statistical mechanics and Brain-Computer, which will benefit both Brain-Computer communities and CVM communities. The manuscript is clearly written with a large number of references, and theoretical description of the CVM is accurate. The referee believes that this manuscript should be published.

Before submitting the final form, however, the author would be recommended to modify/correct the following points.

Since the tutorial introduction of the CVM is given in Chapters 4 and 5 for 1-D system, it is rather easier to grasp the basic concepts of the CVM. Yet, for the discussion of 2-D system, it is desirable to draw a similar figure as Figs. 1-3 to identify the basic cluster. In fact, the basic cluster is the key word to identify the approximation level of the CVM.

In alloy systems it is often observed that h parameter (interaction energy) depends on concentration x (also the local configuration y and z). The author focuses only on analytical solutions in this article, which makes the discussion transparent and clear, however in reality such dependences may not be negligible. Some comments should be offered.  Also, the physical meaning/reality of h parameter in BCIs should be clarified.

It is not easy to understand what are described in the far-right figures in Fig. 1. A bit more detailed explanations should facilitate readers to understand the meaning.

For the future works, the referee sees the importance of extending the present study to the following two directions. One is non-equi-concentration situation. The author can go to numerical calculations and will be able to obtain rich information. The second one is kinetics calculations. Path Probability Method (PPM) was devised by Kikuchi as the natural extension of the CVM to time domain and PPM is quite powerful tool to deal with non-equilibrium process towards the final equilibrium state. The author may give comments if any.

Finally the typos:

line 134;  that can achieved  

line 550; model model

Author Response

These are excellent and highly valuable comments. I greatly appreciate the depth and care that Reviewer 1 has put into the review. I have adopted all the requested minor revisions. Specifically,

Num.

Comment

Action  

Note

1.a

… for the discussion of 2-D system, it is   desirable to draw a similar figure as Figs. 1-3 to identify the basic cluster.

done

Done, along with appropriate Discussions.

1.b

… the basic cluster is the key word to   identify the approximation level of the CVM.

done

Done, see lines 59-60.

2.a

… in reality such dependences may not be   negligible  (of the h parameter on concentration x) … Some   comments should be offered. (w/r/t parameter h)

done

Done by introducing a three new paragraphs (starting with Par 7) in   Section 3 on Statistical Thermodynamics, removing two that were originally   there, and rewriting the transition to the bullet points on “the value of   this for BCI applications …”

2.b

Also, the physical meaning/reality of h parameter in BCIs   should be clarified.

done

Added descriptive paragraph to introduction, lines 83 – 90. Added more description in the Discussion also.

3

It is not easy to understand what are described in the   far-right figures in Fig. 1. A bit more detailed explanations should   facilitate readers to understand the meaning.

done

I am not clear as to whether Reviewer 1 was referring to how the   mottled grey units in the far-right of the patterns corresponded to the wrap-around   from the far left, so have introduced a couple of new sentences in the figure   caption that explain that more. I’ve also introduced a sentence that invites   the reader to the following Section 5 for a discussion of the parameters h,   which are given without explanation in this figure.

4

For the future works, {recommended text here} … The   author may give comments if any.

done

Done. Reviewer 1’s suggested words are inserted towards the end of   the Discussion.

Typo 1

line 134;  that can achieved  

done

Typo 2

line 550; model model

done

Reviewer 2 Report

I enjoyed reading this thorough and interesting account of cluster variation methods and their motivation in relation to systems neuroscience (and brain computer interfacing).  Strategically, your paper is full of really interesting ideas and illustrations; however, it presents no new results or applications.  As such, I think it would be better framed as a primer or tutorial; acknowledging the promissory nature of this work so that readers know what to expect.  This might involve changing the title to something like: “Cluster Variation Methods: A Primer for Neuroscientists”.

To emphasize this point, I think you need to sketch for the reader what they will learn from reading your paper.  I would recommend something like the following:

“The purpose of this paper is to introduce neuroscientists to cluster variation methods and their potential application in systems neuroscience.  This primer focuses on brain computer interfacing; however, the contribution is more wide ranging and can be summarised as follows:  If distributed systems (like the cortex) can be characterized as a two dimensional lattice with local coupling, there exists a set of mathematical tools that enable one to quantify the distribution of local interactions.  In brief, these (cluster variation) tools allow one to prescribe the distribution of local configurations in terms of a single parameter; namely, an interaction enthalpy.  This plays a similar role to the temperature in statistical thermodynamics; thereby prescribing a distribution over configurational variables.  This formulation has a number of potentially exciting applications.  For example, under the assumption that biological systems minimize their free energy, the ensuing equilibrium distribution of local configurations can be determined for any given interaction enthalpy.  This offers a principle and parsimonious way to specify prior probability distributions over the distributed states of (neuronal) networks. These prior distributions may play an important role in either generating random networks for statistical comparison in graph theory – or in furnishing observation or generative models for empirical data via prior constraints.  Another application of this technology would be to summarize any empirical distribution of local configurations in terms of a single parameter enabling one to test for differences between different brain states or, indeed, different subjects and diagnostic categories.  Finally, the ability to summarize the statistics of local configurations on lattices paves the way  for testing key hypotheses about network organization; for example, do real brain networks minimize free energy for a given interaction enthalpy?  We will return to some of these ideas in the discussion.”

In addition, perhaps you could consider the following:

MAJOR POINTS

Your writing style is very clear but a bit staccato.  The text is broken up into too many small paragraphs, which destroys the flow or narrative.  Could I suggest you go through ensuring that no paragraph is less than 8 lines?  This will force you (and the reader) to think about what observations and assertions link to each other.

You introduce a lot of variables.  It would be nice to have a glossary of all variables you use with a brief description.  This would help enormously and avoid having to backtrack to find out what a particular variable means.

The paper is rather long as it stands.  Although the background material on the brain computer interfacing was interesting; it does use up the readers energy.  Unless you have a specific reason for covering brain computer interfacing in such depth, I would reduce this amount of material substantially – this paper should move quickly and gracefully to the key conceptual ideas. 

On page 15 (section 3.4) – I think you can tidy up the links between criticality, complexity and free energy minimization with something like the following:

“From the perspective of the free energy principle (i.e., the minimization of variational free energy), both self-organized criticality and complexity reduction are mandatory.  This follows because the variational free energy can always be expressed as complexity minus accuracy, when the enthalpy turn is interpreted as a log probability density.  This means that minimizing free energy necessarily entails a minimization of complexity.  In this context, complexity is defined as the KL divergence between posterior and prior distributions.  In other words, complexity is the degree to which the distribution is pulled away from the prior (equilibrium) distribution, on exposure to the environment (see Sengupta et al). 

Interestingly, the variational free energy can also be expressed as entropy minus enthalpy.   The entropy term (under some simplifying – Laplacian – assumptions) corresponds to the curvature of the free energy.  This means that a minimum free energy solution necessarily occupies a regime of the free energy that has low curvature.  In turn, this means that local perturbations induce unstable critical or slow fluctuations.  In other words, minimizing free energy necessarily leads to a critical slowing (see Friston et al for details).”

In the discussion, you may also want to add something like:  “In relation to the above applications, if it can be shown that empirical neural network configurations minimize free energy under a given interaction enthalpy, then free energy becomes a Lyapunov function or Lagrangian for neuronal systems.  This may have profound implications for elaborating variational principles of neuronal dynamics; eg gauge fields as discussed in Sengupta et al).”

http://www.ncbi.nlm.nih.gov/pubmed/23935475

http://www.ncbi.nlm.nih.gov/pubmed/22783185

http://www.ncbi.nlm.nih.gov/pubmed/26953636

MINOR POINTS

Page 1 (line 28) – I would say “Bayesian inference; for example approximate Bayesian computation.”

Approximate Bayesian inference is the more general class of procedures - Approximate Bayesian computation is a particular variant usually used in genomics and molecular biology.

Page 2 (line 41) – I would say “of equilibrium stabilized (or non-equilibrium steady state for open systems).”

Page 2 (line 57) – replace “substantial respect” with “substantial attention”.

Page 3 (line 126) – please explain what M =3 functional motif means?

Page 4 (line 234) – replace “can achieved” with “can be achieved”.

Page 7 (line 283) – what is the probability density function over? 

Page 11 (line 497) – did you mean “equilibrium distribution” or “ergodic distribution”?  Ergodic would be more general and would also apply to open systems.

Page 13 (line 563) – replace “greater capabilities” with “greater latitude”.

Figure 1 – why did you use a zigzag example?  Is this a 1D example or a 1D manifold embedded in a 2-space?

Page 17 (line 731) – I think you need a slightly more heuristic introduction into the notion of configuration variables.  Are these the same as the fractional variables and the appendix?  In other words, you might say something like:

“The configurational variables score the frequency with which various local configurations would be sampled from a lattice.  In other words, they are the fraction of times they occur under random sampling.  This means that they can be interpreted in terms of the probability of finding a particular configuration of one or more states when sampling at random or over long periods of time (under ergodicity assumptions).  We will appeal to their interpretation as probabilities later when linking the configuration variables to entropy.”

Page 18 (line 740) – remind the reader what the variables h and z are in heuristic terms, i.e., “These are the first and third……….”

Equation 5 – I would point that this expression is for single units. 

Page 23 and preceding – you have overloaded (used twice) the variable BETA.  This is either a precision or inverse temperature or it is a degeneracy parameter.  My advice is to set BETA =1 at its first introduction and remove this variable from all subsequent Equations and the appendix.  You really do not need the Boltzmann constant either.  This is because, as you say, your formulation is of an abstract sort – not of classical statistical thermodynamics.  I think removing unnecessary constants will greatly simplify your exposition and avoid using too many variables.

Page 28 (line 1019) – you might also mention that a further application would be to ensure that any empirical distribution is indeed a minimum of a free energy functional.  This would be an important contribution and suggest that a variational principle of free energy minimization is actually in play for the system understudy. 

I hope that these comments help should any revision be required.

Author Response

An excellent set of suggestions, most gratefully adopted and used in entirety.

Reviewer 2 Comments

Num.

Comment

Action

Note

1.a

… I think it would be better   framed as a primer or tutorial; acknowledging the promissory nature of this   work so that readers know what to expect. 

done

Both the abstract and the introduction (within the first few   paragraphs) have been changed to reflect this emphasis.

The abstract is still at 200 words. (See extracted Abstract at the   end of this paper; 200 word count precisely.)    A set of enumerated points highlighting the value of this paper for   the brain research and BCI communities has been brought forward into the   Introduction from its previous Background location.

1.b

… changing the title to something   like: “Cluster Variation Methods: A Primer for Neuroscientists”.

done

The title has been changed to “The Cluster Variation Method: A Primer   for Neuroscientists”

2

I think you need to sketch for the reader what they will learn from   reading your paper.  I would recommend something like the following: (suggested   paragraph follows)

done

Reviewer 2 suggested an excellent opening paragraph, which I have   used nearly as written by Reviewer 2 (small modifications to address the   mathematical specifics).  See this as   the new starting paragraph for the article.  

3

.    The text is broken up into too many small paragraphs, which destroys the flow   or narrative.  Could I suggest you go through ensuring that no paragraph   is less than 8 lines? 

done

I have rewritten and edited Sections 1 – 3 (which is where the “broken   up” text elements occurred) in order to achieve better flow and also reduce   the amount of text used.

4

It   would be nice to have a glossary of all variables you use with a brief   description. 

done

Included as Table 1 in Section 3.

5

Unless   you have a specific reason for covering brain computer interfacing in such   depth, I would reduce this amount of material substantially – this paper   should move quickly and gracefully to the key conceptual ideas. 

done

I have significantly reduced the amount of text devoted to the   pragmatics of BCIs, and substantially condensed Section 2, “Background.”

6

On page 15 (section 3.4)   – I think you can tidy up the links between criticality, complexity and free   energy minimization with something like the following: (suggested paragraph   follows)

done

Paragraph included per Reviewer 2’s recommendations, with a few   references included that will help the reader trace this particular line of   thought.

AJM: NEED to go back and   carefully reread and consider!

7

In the discussion, you   may also want to add something like: (recommended paragraph follows,   together with suggested references)

done

Done, see end of the Discussion. Ref. included to Sengupta et al.   (2016), “Towards a neuronal gauge theory.”

8

Page 1 (line 28) – I   would say “Bayesian inference; for example approximate Bayesian computation.”

done

9

Page 2 (line 41) – I   would say “of equilibrium stabilized (or non-equilibrium steady state for   open systems).”

done

10

Page 2 (line 57) –   replace “substantial respect” with “substantial attention”.

done

11

Page 3 (line 126) –   please explain what M =3 functional motif means?

done

The material previously at line 126 has been moved up to line 52.

A brief phrase giving the meaning of “M=3 functional motif” has been   included in the enumerated point at line 45. The original paragraph has been   expanded with additional phrasing to describe the meaning of the “M=3   functional motif” in lines 52-56.

While this author thanks Reviewer 2 for suggesting a bit of   embellishment, the term “functional motif” is well-known in the brain   research community, and a quick glance at Ref. 10 (cited at the end of that   paragraph) would immediately take the reader to a highly-regarded work that   explains functional and structural motifs in great detail.

12

Page 4 (line 134) –   replace “can achieved” with “can be achieved”.

done

Same as Typo 1 as noted by Reviewer 1

13

Page 7 (line 283) – what   is the probability density function over? 

done

Addressed by including more detail from the referenced works.

14

Page 11 (line 497) – did   you mean “equilibrium distribution” or “ergodic distribution”?  Ergodic   would be more general and would also apply to open systems.

done

Done, this is in Sect. 3, nine paragraphs from the beginning of the   Section, and I’ve added a brief phrase describing what it means to be “ergodic,”   as neuroscientists are not likely to be familiar with the term.

15

Page 13 (line 563) –   replace “greater capabilities” with “greater latitude”.

done

16

Figure 1 – why did you   use a zigzag example?  Is this a 1D example or a 1D manifold embedded in   a 2-space?

done

This is a 1-D example, and is explained further now in Subsection   4.2, , “Introducing the Configuration Variables.”

17

Page 17 (line 731) – I   think you need a slightly more heuristic introduction into the notion of   configuration variables.  Are these the same as the fractional variables   and the appendix?  In other words, you might say something like:   (suggested paragraph follows)

done

The author has changed the term “fractional variables” in the   Appendix to “configuration variables,” and has included the Reviewer’s   recommended paragraph that helps explain the nature of configuration   variables at the beginning of Sect. 4., “Configuration Variables: Describing   Local Patterns.”

18

Page 18 (line 740) –   remind the reader what the variables h and z are in heuristic terms, i.e.,   “These are the first and third……….”

done

Done, see parenthetical notes inserted into Par. 6 in Sect. 4.3.   (Four paragraphs after the bullet points on patterns A, B, and C.)

19

Equation 5 – I would   point that this expression is for single units. 

done

20

Page 23 and preceding –   you have overloaded (used twice) the variable BETA.  This is either a   precision or inverse temperature or it is a degeneracy parameter.  My   advice is to set BETA =1 at its first introduction and remove this variable   from all subsequent Equations and the appendix.  You really do not need   the Boltzmann constant either.  This is because, as you say, your   formulation is of an abstract sort – not of classical statistical   thermodynamics.  I think removing unnecessary constants will greatly   simplify your exposition and avoid using too many variables.

done

I have removed the term BETA from all equations except those few   instances where BETA(subscripted) refers to a degeneracy factor. I have   reformulated the equations with the constant of BETA = 1/(kT) set to 1. I   have adjusted the figures accordingly.

21

Page 28 (line 1019) – you   might also mention that a further application would be to ensure that any   empirical distribution is indeed a minimum of a free energy functional.    This would be an important contribution and suggest that a variational   principle of free energy minimization is actually in play for the system   under study. 

done

Reviewer 1’s recommending wording has been introduced into the second   set of enumerated points of the Discussion (see point 2 of the three points,

Brain Sci. EISSN 2076-3425 Published by MDPI AG, Basel, Switzerland RSS E-Mail Table of Contents Alert
Back to Top